# How riparian and floodplain restoration modify the effects of increasing temperature on adult salmon spawner abundance in the Chehalis River, WA

**Caleb B. Fogel** [1]*, **Colin L. Nicol** [1], **Jeffrey C. Jorgensen** [2], **Timothy J. Beechie** [2], **Britta Timpane-Padgham** [3], **Peter Kiffney** [2], **Gustav Seixas** [4], **John Winkowski** [5]

**1** Ocean Associates, Inc., Seattle, Washington, United States of America, **2** National Oceanic and Atmospheric Administration, National Marine Fisheries Service, Northwest Fisheries Science Center, Fish Ecology Division, Seattle, Washington, United States of America, **3** A.I.S, Inc., Seattle, Washington, United States of America, **4** Skagit River System Cooperative, Burlington, Washington, United States of America, **5** Washington Department of Fish and Wildlife, Olympia, Washington, United States of America

\* caleb.fogel@gmail.com

**Data Availability Statement:** Model code can be found at the following repository: https://zenodo.org/record/6590079#.YpJYJBPMJpQ. GIS files can

## Abstract

Stream temperatures in the Pacific Northwest are projected to increase with climate change, placing additional stress on cold-water salmonids. We modeled the potential impact of increased stream temperatures on four anadromous salmonid populations in the Chehalis River Basin (spring-run and fall-run Chinook salmon *Oncorhynchus tshawytscha*, coho salmon *O. kisutch*, and steelhead *O. mykiss*), as well as the potential for floodplain reconnection and stream shade restoration to offset the effects of future temperature increases. In the Chehalis River Basin, peak summer stream temperatures are predicted to increase by as much as 3˚C by late-century, but restoration actions can locally decrease temperatures by as much as 6˚C. On average, however, basin-wide average stream temperatures are expected to increase because most reaches have low temperature reduction potential for either restoration action relative to climate change. Results from the life cycle models indicated that, without restoration actions, increased summer temperatures are likely to produce significant declines in spawner abundance by late-century for coho (-29%), steelhead (-34%), and spring-run Chinook salmon (-95%), and smaller decreases for fall-run Chinook salmon (-17%). Restoration actions reduced these declines in all cases, although model results suggest that temperature restoration alone may not fully mitigate effects of future temperature increases. Notably, floodplain reconnection provided a greater benefit than riparian restoration for steelhead and both Chinook salmon populations, but riparian restoration provided a greater benefit for coho. This pattern emerged because coho salmon tend to spawn and rear in smaller streams where shade restoration has a larger effect on stream temperature, whereas Chinook and steelhead tend to occupy larger rivers where temperatures are more influenced by floodplain connectivity. Spring-run Chinook salmon are the only population for which peak temperatures affect adult prespawn survival in addition to rearing survival, making them the most sensitive species to increasing stream temperatures.

be found at the following link: https://www.
fisheries.noaa.gov/resource/tool-app/chehalis-
watershed-assessment-and-salmon-life-cycle-
modeling.

**Funding:** This work was supported by the
Washington Department of Fish and Wildlife under
contract WDFW #15-03970, and by the
Washington Recreation and Conservation Office
under contract RCO #17-1477. WDFW assisted
with data collection and provided local expert
knowledge of the study system, but played no role
in the decision to publish.

**Competing interests:** The authors have declared
that no competing interests exist.

## Introduction

Anthropogenic climate change is expected to negatively affect species worldwide, leading
to high extinction risk across a range of taxa and continents [1, 2]. Climate change
projections include warming global mean air and ocean temperaturesshifts in global precipita-
tion patterns, and increases in regional risk of drought and the occurrence and intensity of
extreme weather events [3, 4]. Freshwater ecosystems in particular are expected to experience
significant changes in environmental conditions, such as flow and water temperature
regimes, that could limit habitat area and quality for many native species [5–7]. Given the
threat that human activity poses to many freshwater species, it is essential to understand the
impacts that climate change may have on habitat quality in order to effectively manage resil-
ient ecosystems.

Populations of anadromous salmonids (*Oncorhynchus* spp.) in the Pacific Northwest are
significantly diminished from historical levels, largely due to direct and indirect effects of
human activities and land use practices [8–11]. Anadromous salmonids are cold-water
adapted species that spend a portion of their lives within freshwater habitat. The projected
effects of climate change pose substantial risk to salmonids. These effects include changes
to flow regimes resulting from altered patterns in precipitation and snowmelt [12],
decreased habitat area due to increased drought, and decreased habitat quality due to
warmer than optimum stream temperatures [13]. Depending on food availability, increased
temperatures are expected to alter summer growth and mortality rates of salmonids, which
will likely lead to declines in native populations [14, 15]. However, in some cases warmer
stream temperatures outside of summer months may also provide some benefit to salmo-
nids at certain life stages, such as the juvenile stage, by increasing their growth potential
[16].

While climate change models predict substantial changes to salmonid freshwater habitats
[14, 17], there is potential to offset some habitat changes or population declines through a vari-
ety of restoration actions [18, 19]. Some forms of restoration may also increase species resil-
ience to climate change [20]. However, few studies quantify the thermal benefits of riparian
restoration and increased floodplain connectivity in terms of their effect on adult salmon pop-
ulations (but see, e.g., [21]). Restoration of the riparian canopy in deforested reaches may
reduce stream temperatures by intercepting incoming solar radiation [22–28], while reconnec-
tion of hyporheic flow paths through increased floodplain connectivity may increase inputs of
cooler water to streams [29, 30]. Projected stream temperature increases in the Chehalis River
Basin in SW Washington, USA, are likely to threaten the viability of salmonid populations by
the end of the 21st century unless restoration actions can, in part, alleviate these increases [25].
Current temperatures in the basin already exceed critical temperature thresholds in some
areas during summer [31], and the spatial extent of these sub-optimal water temperatures is
likely to expand in the coming decades [14, 31, 32].

In this paper, we use a series of quantitative tools to estimate the effect of future increases in
stream temperature due to climate change on anadromous Pacific salmonid populations in the
Chehalis River Basin. We focus on the effectiveness of two restoration actions aimed at
decreasing stream temperatures: encouraging hyporheic flow through floodplain reconnection
[33, 34], and increasing stream shading through planting and tree growth within the riparian
corridor. Our study builds upon recent studies that explore restoration potential for salmonid
populations within the Chehalis River Basin by adding a climate change component to the
Habitat Assessment and Restoration Planning (HARP) model described in Beechie et al. [35]
and Jorgensen et al. [36].

## Materials and methods

### Study area and species

The Chehalis River Basin is the second largest watershed in the state of Washington, with a drainage area of approximately 6,900 km$^2$ and a total main stem length of 185 km. The basin includes portions of the Willapa Hills, Cascade Mountains, and Olympic Mountains, and empties into the Pacific Ocean at Grays Harbor near the town of Aberdeen (Fig 1). The Chehalis River basin consists of 63 subbasins and 10 Ecological Regions as defined by the Chehalis Basin Scientific Review Team (S1 Fig) (See [35, 36]). The hydrograph for the basin is rainfall-dominated [37]. A very small portion of the basin in the Olympic Mountains consists of higher elevation tributaries where stream flows are dominated by snowmelt; however, future changes to snowpack from climate change are unlikely to impact flow or temperature regimes throughout the majority of the basin. Recent estimates of the maximum seven day average of daily maximum stream temperatures for August (7-DADM) range from ~7˚C in some tributaries to > 27˚C in portions of the mainstem Chehalis River [31, 32]. Land use is predominantly managed forest, with agricultural and urban areas concentrated in lower elevation floodplains (Fig 1) [38]. Current riparian shade levels are reduced in many agricultural and urban areas, whereas riparian areas within managed forests have more shade due to riparian protections that have been in place for over 20 years (Washington State Salmon Recovery Act was passed in 1999; subsequent administrative rules were enacted in 2001).

Species composition of riparian forests in the study area varies as a function of geomorphic setting [35, 39], so we stratified rivers into two types following Seixas et al. [25]: non-floodplain channels, which have stable riparian landforms (terraces or hill slopes), and floodplain channels, which have varying rates of lateral channel migration and floodplain turnover. Non-floodplain riparian areas are typically dominated by mature conifer species under natural conditions [40, 41] with an average height of ~52 m [25], whereas floodplain forests tend to have a mix of coniferous and deciduous species [39, 42] and an estimated average tree height of ~30 m [25]. We used these tree heights as the natural potential (mature) riparian condition for estimating future temperatures in the restoration scenarios.

Our analysis focuses on four populations of anadromous salmonids: spring-run and fall-run Chinook salmon (*Oncorhynchus tshawytscha*), coho salmon (*O. kisutch*), and steelhead (*O. mykiss*) (see S.1 for spawning and rearing distribution). Because the timing of life stages varies between populations, so does the effect of temperature on individual life stages. Spring-run Chinook salmon are the only population with adult prespawn holding in summer and therefore are the only population vulnerable to exposure to high temperatures during this life stage [43, 44]. Juvenile coho salmon and steelhead rear in freshwater habitats throughout the entire summer, and consequently are exposed to the highest annual stream temperatures as juveniles [43, 45, 46]. Steelhead spend up to three summers in fresh water, with most juveniles leaving fresh water after two summers [36, 47, 48]. Most juvenile spring-run and fall-run Chinook salmon have migrated out of freshwater habitats by the time streams reach their highest temperatures, however the remaining juveniles are exposed to warming temperatures during outmigration in June [46, 48]. Sensitivities to temperature vary among species and life stages, although the Upper Incipient Lethal Temperature (UILT) for juvenile salmonids is generally in the range of 24–26˚C [49–51].

### Temperature and habitat modeling framework

We created five temperature scenarios to explore the potential effects of projected future summer stream temperatures on the four salmon and steelhead populations (Table 1). Each future

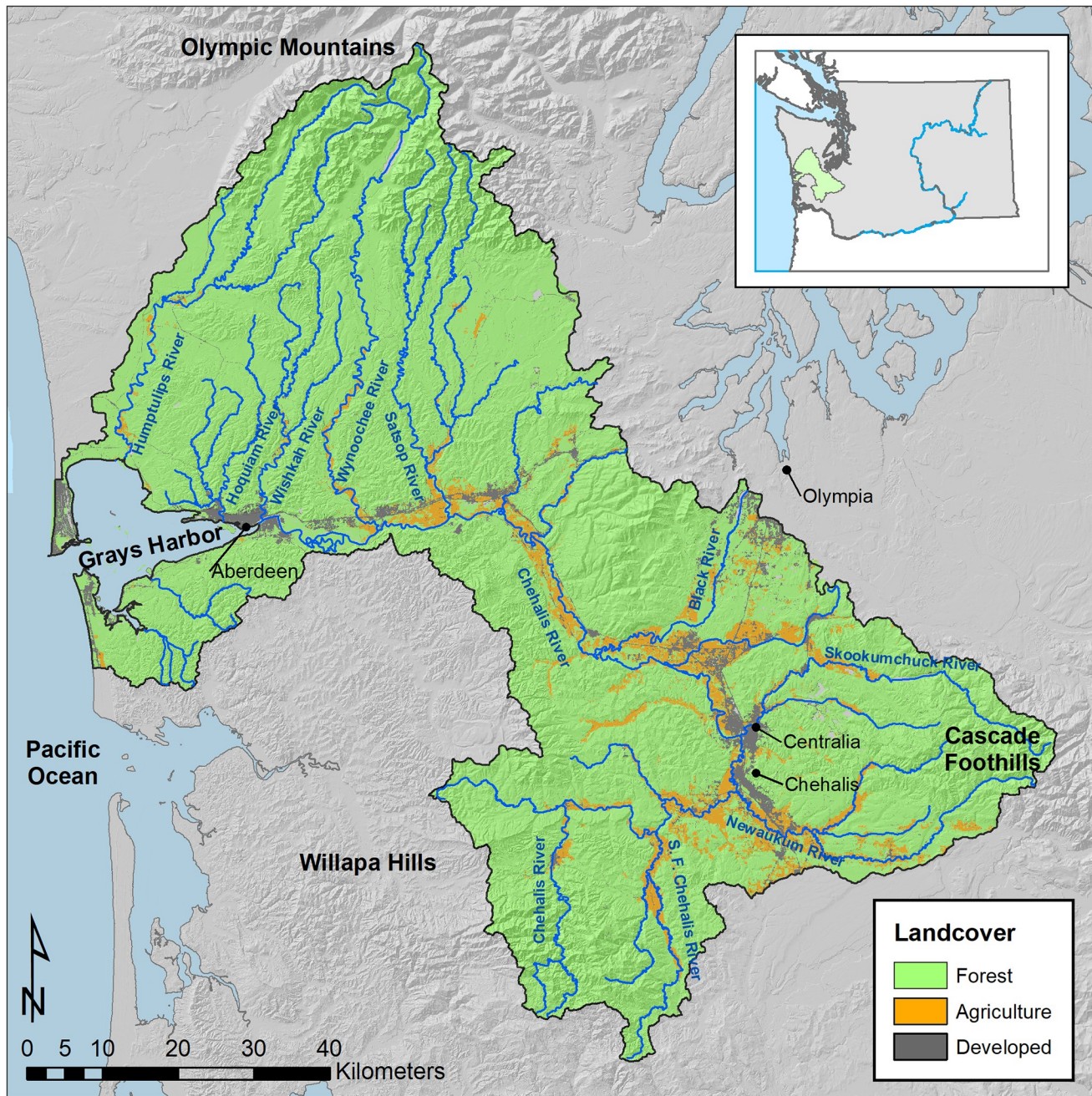

**Fig 1. Study area.** Map of the Chehalis River Basin showing major tributaries, cities, regions, and land use categories. Forest lands are primarily managed industrial forests but include portions of the Olympic National Forest and Olympic National Park. Inset map shows the location of the Chehalis River Basin (green) within the state of Washington, and in relation to the Columbia River (blue).

temperature scenario was modeled for both mid-century (2040s) and late-century (2080s) conditions. We first modeled current conditions as a baseline against which to compare future conditions. We then modeled one no-action scenario in which temperatures increased due to climate change but no restoration actions were taken. Finally, we modeled three restoration scenarios that assessed the potential for mitigation of climate change impacts on temperature through targeted restoration actions. The restoration scenarios included tree planting and

**Table 1. Description of temperature scenarios.** See text for hypotheses for each scenario.

| Temperature Scenario | Description |
| --- | --- |
| Current | Current temperatures (Washington Dept. Fish & Wildlife Thermalscape) |
| No-action | Future temperatures with climate change effects and no riparian restoration or floodplain reconnection |
| Riparian | Future temperatures with climate change and riparian restoration (tree planting and protection of existing riparian corridor) |
| Floodplain | Future temperatures with climate change and floodplain reconnection |
| Combined | Future temperatures with climate change, riparian restoration, and floodplain reconnection |

protection within riparian areas to increase stream shading; floodplain reconnection to cool rivers by increasing hyporheic flow; and a combined scenario with both riparian restoration, and floodplain reconnection.

We used the HARP model described in Beechie et al. [35] and Jorgensen et al. [36] to evaluate the five temperature scenarios. The HARP model is a framework for evaluating constraints on salmon populations due to changes in habitat conditions, with the primary purpose of comparing the potential benefit of alternative habitat restoration actions (Fig 2). This multi-step analysis begins with an assessment of habitat changes from historical to current conditions, which defines the restoration potential for habitat restoration options such as migration barrier removal, fine sediment reduction, wood augmentation, shade restoration, floodplain reconnection, or beaver restoration, under current climate conditions [35]. Next, habitat conditions under the historical and current time periods are used to estimate life stage- and species-specific capacities and productivities for each species and subbasin, which are then incorporated into salmon life cycle models. Finally, the life-cycle models estimate the effect of individual or multiple habitat changes on salmon populations. Those analyses highlighted specific restoration actions that are most likely to increase spawner abundance for each species under current climate conditions. For example, coho salmon were likely to benefit most from floodplain reconnection and beaver restoration, whereas spring-run Chinook salmon were likely to benefit most from floodplain reconnection, shade restoration, and wood augmentation [36].

In this paper we use the same model framework to evaluate changes in habitat from present to future conditions by incorporating climate change projections and restoration actions, focusing on temperature change (Fig 2). We modified the life-stage specific fish capacities and productivities based on stream temperature conditions for each scenario, and then used these estimates as inputs to life-cycle models to estimate changes in spawner abundance. We used the August 7-DADM to describe temperature change from current conditions under each scenario. Our modeled restoration scenarios assumed that restoration would occur in all reaches with restoration potential (i.e. reaches with degraded riparian condition or disconnected floodplain habitat).

## Temperature estimates

**Current temperatures.** We used current temperature estimates from the Washington Department of Fish and Wildlife (WDFW) Chehalis Thermalscape spatial stream network temperature model [31, 32] for the current scenario in our model. The Chehalis Thermalscape model estimates spatially continuous August average of daily average stream temperatures (August ADA) for the entire Chehalis River Basin at intervals of approximately 500–1000 m. Temperature estimates were modeled using measured stream temperatures from over 120

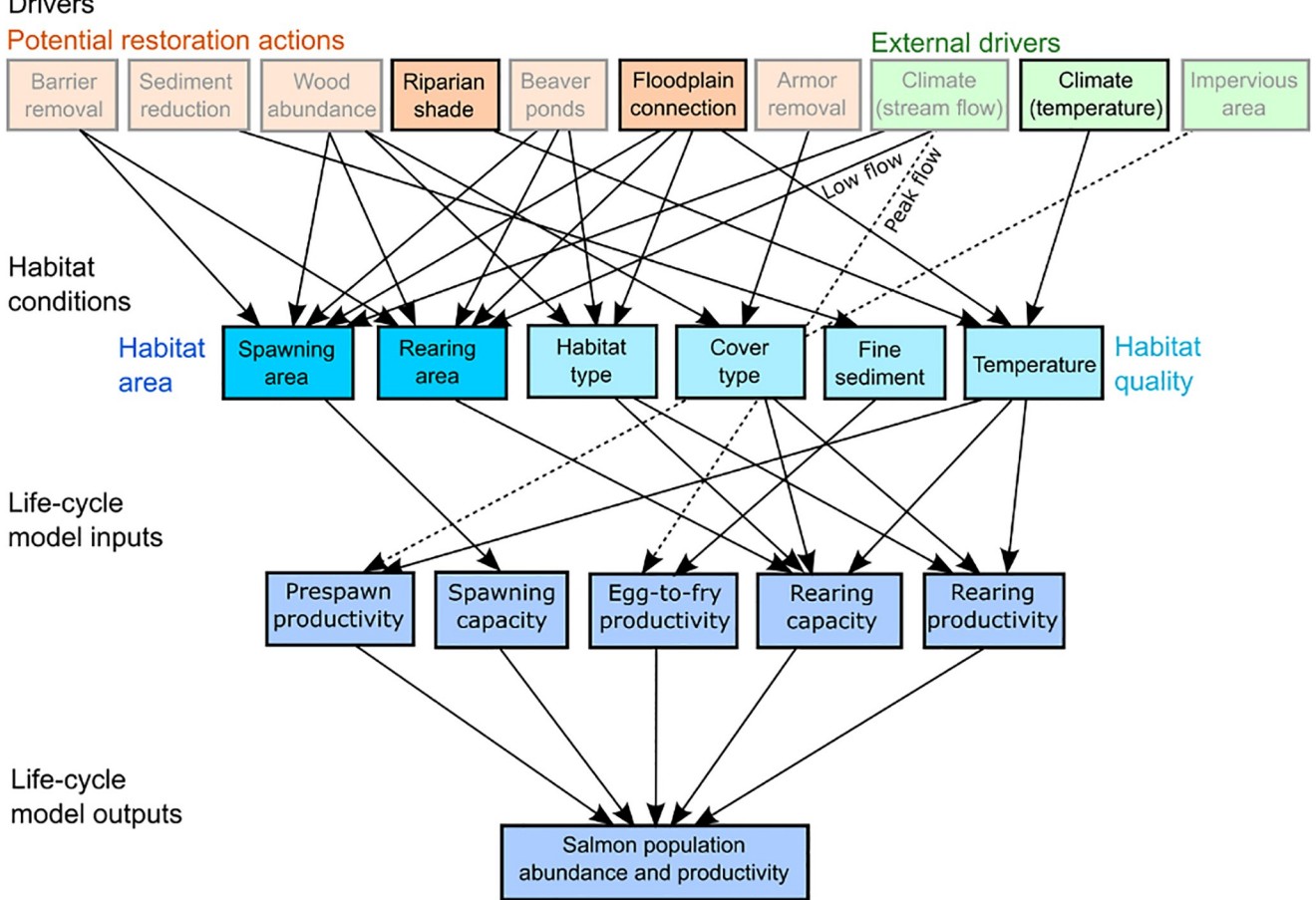

**Fig 2. HARP conceptual model.** Diagram of model linkages between restoration actions and external drivers, habitat conditions, life-cycle model input parameters, and salmon population response. Restoration actions and external drivers included in this assessment were Riparian shade, Floodplain connection, and Climate (temperature).

sites throughout the basin, and a series of temporal (August air temperature, mean August stream flow) and spatial (elevation, stream slope, annual precipitation, drainage area, riparian canopy cover, base-flow index, lake percentage) covariates. For our analysis we segmented the Chehalis Basin river network into reaches approximately 200 m in length, in order to assess a range of habitat conditions at high resolution [35, 36]. Reaches were assigned baseline temperatures from the nearest Chehalis Thermalscape data point, or from the average of the nearest temperature data points if a reach was equidistant from multiple data points.

The temperature-survival functions used in our life cycle model require temperature metrics that are specific to the timing of life stages sensitive to temperature. Coho and steelhead summer rearing, and spring-run Chinook salmon pre-spawning occur throughout the summer [43–46], whereas a large percentage of the total spring-run and fall-run Chinook salmon outmigration occurs between June 1 and June 21 [46, 48]. Therefore, we used the August 7-DADM in our coho and steelhead summer rearing, and spring-run Chinook salmon pre-spawning temperature-survival functions, and the June 1–21 average of daily maximum temperatures (June 1–21 ADM) for the Chinook salmon outmigration temperature-survival function. To convert August ADA estimates from the Chehalis Thermalscape dataset to 7-DADM and June 1–21 ADM, we used measured temperatures from the same temperature sensor sites

used in the Chehalis Thermalscape model [31]. We calculated the August ADA (n = 80 sites), the 7-DADM (n = 80 sites) and the June 1–21 ADM (n = 43 sites) at each site, and developed conversion equations by regressing the 7-DADM and June 1–21 ADM against the August ADA (S2 Fig).

**Future no-action (climate change) temperatures.**   We used the future stream temperature projections from the Chehalis Thermalscape model in our future climate change scenarios. The Chehalis Thermalscape model estimates mid-century and late-century stream temperatures by incorporating projected changes to air temperature and stream flow [31, 32]. Air temperature estimates were based on the IPCC A1B emissions scenario for Western Washington [31, 32]. The A1B scenario reflects continued economic growth with a global population peak by mid-century, and a balanced use of fossil and non-fossil energy sources, and was the ensemble average of 10 global climate models (GCMs) [52]. We do not yet have stream temperature projections based on the newer IPCC AR5 scenarios for RCP 4.5 and 8.5, which project a peak in $CO_2$ emissions by mid-century followed by a decline, and continued increases in $CO_2$ emissions respectively [53]. Future stream flow estimates used in the Chehalis Thermalscape model were calculated using the Variable Infiltration Capacity (VIC) hydrologic model [54] along with conditions from the A1B scenario (see [17]), and time series flow data from three stream gages within the Chehalis River Basin. Projected future increases in stream temperature, converted in our model to 7-DADM, were +1.15°C by mid-century and +3°C by late-century, and did not vary significantly among reaches at the scale of the Chehalis River Basin (range < 0.0001°C across all reaches) [31, 32].

**Future temperature change—Stream shading.**   We used a tree growth model to estimate future shade, and the Seixas et al. [25] stream temperature model to estimate future temperature change from stream shading. The Seixas et al. model was less accurate at absolute temperature prediction than the Chehalis Thermalscape model, so we used the Chehalis Thermalscape model to predict current temperature. However, the Chehalis Thermalscape model cannot predict change in stream temperature due to changes in shade, so we extracted the shade component of the Seixas et al. model to estimate change from current temperature due to tree growth.

We first estimated canopy opening angle (a correlate of shade) under current conditions for each 200-m reach using recent Light Detection and Ranging (lidar) data from the Puget Sound Lidar Consortium, and aerial imagery. Where lidar data were available, we used an automated process to estimate the canopy opening width and the tree height on each side of the channel from the lidar data [25]. Tree height was calculated from the lidar dataset as the difference between the elevations of the first returns and the ground surface. Where lidar data were unavailable, we visually estimated canopy opening width and tree size class (Table 2) on each bank at approximately 200m intervals using recent aerial imagery [35]. We then assigned these values to our segmented 200 m reaches based on proximity.

To estimate the change in tree height by mid- and late-century, we used growth rates of conifer and deciduous trees for the Pacific Northwest region [55, 56]. Growth rates for both tree types were rapid at first and leveled off with increasing age. Although future changes to air

**Table 2. Median lidar tree height in each size class.**

| Height Class | Median Tree Height |
| --- | --- |
| Tall | 34 m |
| Medium | 23 m |
| Short | 14 m |
| No Vegetation | 0 m |

temperature and precipitation patterns may impact tree growth rates, the rates used in our study did not vary with differences in air temperature or other factors. Estimated changes in tree height were added to current tree heights to predict future heights for both mid- and late-century. Canopy opening width was held constant from current conditions and canopy opening angle was recalculated using the updated future tree heights.

Winkowski and Zimmerman [31, 32] examined a number of potential predictor variables including forest cover for the Chehalis Thermalscape model, however, only mean August air temperature, mean August discharge, elevation, and mean annual precipitation were significant predictors [31, 32]. Therefore, to account for temperature changes due to shade, we extracted the canopy opening angle effect of the Seixas model and used that to forecast or hindcast temperature change due to differences in riparian tree height. That is, the change in temperature due to change in canopy opening angle was calculated from the Seixas et al. model as:

$$\Delta T_{(growth)} \; = \; T_{7DADM\,(growth)} \; - \; T_{7DADM\,(current)}$$

Here, *growth* represents future temperature resulting from tree growth only, and does not include climate change impacts. For a given reach, this is equivalent to rearranging the Seixas et al. temperature equation so that:

$$\Delta T_{(growth)} \; = \; 0.035(\Delta\theta)$$

Where $\Delta T$ is the change in stream temperature and $\Delta\theta$ is the change in canopy opening angle due to change in tree height. We added $\Delta T_{(growth)}$ to the current temperatures from the Chehalis Thermalscape dataset in order to calculate future temperatures resulting from tree growth for the riparian and combined scenarios. In reaches with little or no existing riparian vegetation, we assumed that tree planting and subsequent tree growth would occur under the riparian and combined scenarios.

**Future temperature change—Floodplain reconnection.** Connected floodplains can have high rates of hyporheic exchange through gravel bars and the floodplain. These flows can reduce stream temperature and create local thermal refuges during summer [29, 57, 58]. One study on the Willamette River projected that reconnecting floodplain features within a floodplain corridor roughly 425 m wide would decrease 7-DADM temperature by 2˚C [30]. To model alternative restoration scenarios, Seedang et al. [30] created a linear model, relating temperature reduction to the area of connected floodplain features (primarily channels and islands), despite underlying complexities in the estimation of flow path length, hydraulic head, cooling in each flow path, and other factors. While one might expect that temperature reduction will decrease as reconnected floodplains become very wide, we are unaware of other models that may describe a non-linear relationship. Moreover, the floodplain reconnection widths we modeled are narrower than the maximum width modeled in our source study, so we are applying the linear model within a floodplain width range that is consistent with the original model [30].

Because proposed floodplain restoration actions in the Chehalis River Basin vary with channel width, we scaled the 2˚C projection to each channel width and proposed floodplain width (Table 3). We first estimated the widths of floodplain corridors for each reach based on bankfull width using a relationship outlined in the Aquatic Species Restoration Plan (ASRP) guidelines for the Chehalis River Basin (Table 3). We then applied the linear relationship between restored floodplain width and stream temperature change [30] (Table 3) to estimate temperature change to all large river reaches (> 20 m bankfull width) throughout the basin, as well as any small stream reaches (< 20 m bankfull width) with documented disconnected marsh habitat. This assumes that all large river reaches in the basin have some amount of disconnected floodplain habitat, and that restoration would reconnect the planned floodplain width

**Table 3. Relationship between river class, floodplain width, and temperature change for the floodplain restoration scenario.**

| River class | Bankfull width | Width of connected floodplain | Temperature change (°C) |
|---|---|---|---|
| Large River (> 20m bankfull) | > 30m | 305m | -1.43 |
| | 20–30m | 213m | -1 |
| Small Stream (< 20m bankfull) | 10–20m | 152m | -.72 |
| | < 10m | 61m | -.29 |

(Table 3). While historical floodplain widths often exceed those in the restoration plan, we did not model a temperature effect of wider (historical) floodplains because we were not certain that temperature reduction continues as floodplain width increases beyond those widths of the original study. Hence, the modeled temperature reductions in Table 3 also represent the maximum potential temperature reduction for floodplain reconnection in the model. After calculating change in 7-DADM, we converted the modeled change in 7-DADM to a change in June 1–21 ADM by regressing the June 1–21 ADM against the 7-DADM using the measured daily temperatures from the same sites used in the Chehalis Thermalscape model:

$$\Delta T_{June1-21\ ADM} \ = \ 0.98\,(\Delta T_{7-DADM})$$

## Modeling effects of temperature change on spawner abundance

The effects of stream temperature change on the four salmon and steelhead populations, as measured by changes in spawner abundance, were modeled using the HARP life cycle modeling approach described in Jorgensen et al. [36]. Temperature influences habitat quality, which impacts both density independent survival (productivity) and capacity, defined as the number of individuals that can be sustained by a particular habitat. We modeled the effect of temperature on the following life stages: spring-run Chinook salmon upstream migration, spring- and fall-run Chinook salmon sub-yearling outmigration, and coho and steelhead summer rearing (Table 4). For each species and life stage impacted by temperature, we developed a temperature multiplier based on documented relationships between stream temperatures and salmonid capacity and productivity. We then used these multipliers to scale the capacity and productivity parameters used in the life cycle models (described below).

For the current condition (baseline) scenario, we used estimates of current habitat area, rearing and redd density, and life stage productivity for all habitat types (i.e., riffle, pool, beaver pond, bank edge, armored bank edge, backwater, mid-channel, marsh, pond, lake, slough, side-channel pool, side-channel riffle) from Beechie et al. [35] and Jorgensen et al. [36]. Estimates of current habitat area and condition for each subpopulation were obtained through geospatial analysis and modeling [35], whereas density and productivity estimates were drawn

**Table 4. Life stages used in life cycle model [36].** Bolded stages experience some impact from high summer stream temperature.

| Run | Life stages modeled |
|---|---|
| Spring Chinook | **Upstream migration**, spawning, egg incubation, fry colonization, **subyearling rearing** |
| Fall Chinook | Upstream migration, spawning, egg incubation, fry colonization, **subyearling rearing** |
| Coho | Upstream migration, spawning, egg incubation, fry colonization, **summer rearing**, winter rearing |
| Steelhead | Upstream migration, spawning, egg incubation, fry colonization, **ages 0–2 summer rearing**, ages 0–2 winter rearing |

from field surveys and values from published literature (see Jorgensen et al. [36]). Life stage capacity was calculated as the product of habitat area and life stage- and species-specific density [36]. For future scenarios, we applied capacity and productivity multipliers based on stream temperatures to simulate the impacts of temperature on habitat quality. As described previously, we used future estimates of temperature change due to climate change and restoration of riparian shade or floodplain connectivity to predict future stream temperatures under each scenario. In Beechie et al. [35] and Jorgensen et al. [36], floodplain reconnection had the additional benefit of opening up new high-quality habitat area to salmon and steelhead for rearing and spawning. However, the goal of this paper is to address potential actions to mitigate increased stream temperature; therefore, we focus solely on the effects of floodplain reconnection on stream temperature and do not consider the effects of floodplain reconnection on available area for spawning and rearing.

**Temperature multipliers.** We used the following function to calculate the temperature multiplier applied to juvenile coho summer rearing capacity and productivity (Fig 3):

$$Temp\_multiplier_{coho} = \begin{cases} 1 & if\ T < 17°C \\ 0.09\,T + 2.55 & if\ 17°C \leq T < 28°C \\ 0 & if\ T \geq 28°C \end{cases}$$

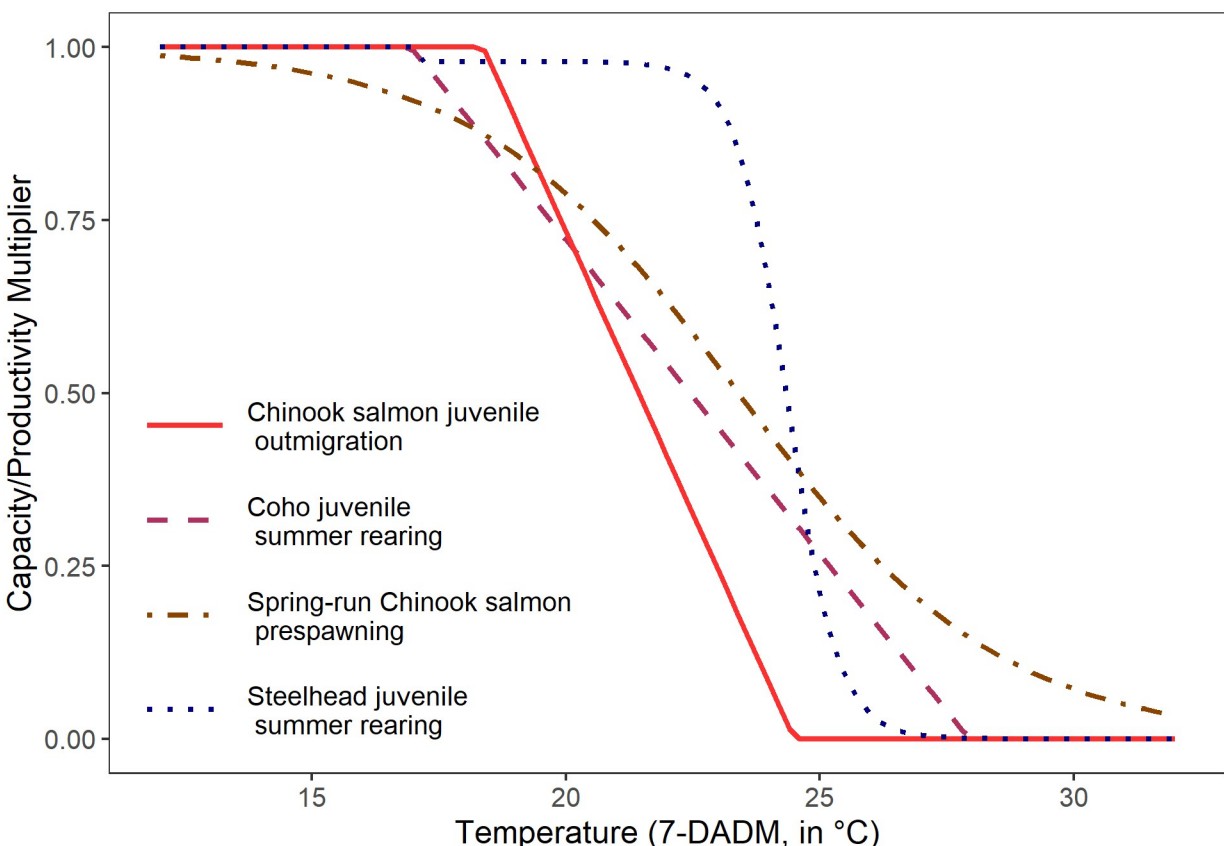

**Fig 3. Functional relationship between the temperature multiplier and the August maximum seven day average of daily maximum temperatures (7-DADM, ˚C) for Chinook salmon juvenile outmigration [43] (see [60, 61]), coho juvenile summer rearing (adapted from [51, 59]), spring-run Chinook salmon prespawning (adapted from [62]), and steelhead juvenile summer rearing [63].**

Where $T$ is the 7-DADM stream temperature and $Temp\_multiplier_{coho}$ is the temperature multiplier. This function was developed using ideal growth range and Critical Thermal Maximum data for coho salmon [51]. We chose the lower limit of our function to be 17˚C, which is the upper end of the ideal temperature range for juvenile coho fed maximum rations [51, 59]. The upper end of our function, 28˚C, represents the lower end of the Critical Thermal Maximum for coho [51]. We assumed the temperature multiplier to be 1 at temperatures <17˚C, to decrease linearly from 1 to 0 in the 17˚C to 28˚C range, and to be 0 at temperatures >28˚C.

We used an experimentally-derived relationship between juvenile steelhead productivity and stream temperature to calculate the temperature multiplier for juvenile steelhead summer rearing capacity and productivity (Fig 3) [63]. In developing this relationship, juvenile steelhead were exposed to temperatures ranging from 8˚C to 30˚C in two-degree increments and mortality was recorded for each trial. The resulting regression function was:

$$Temp\_multiplier_{steelhead} = \frac{97.88}{1 - e^{-((T-24.3522)/-0.5033)}}$$

where $T$ is the 7-DADM and $Temp\_multiplier_{steelhead}$ is the temperature multiplier.

We applied the same juvenile outmigration capacity and productivity temperature multipliers for both spring-run and fall-run Chinook salmon (Fig 3). We used the June 1–21 time period to estimate the temperature effect on the outmigration productivity of juvenile Chinook salmon because this is the period of peak outmigration for Chinook salmon in the Chehalis River Basin [48]. Although juvenile Chinook salmon in the Chehalis river can outmigrate from early spring to mid-summer, 45% of parr outmigrate between June 1 and June 21 [48]. Therefore, the temperature multiplier for Chinook salmon outmigration is applied to only 45% of the population. The functional relationship between the June 1–21 ADM and the productivity multiplier for spring-run and fall-run Chinook salmon outmigration, defined in the 2014 Aquatic Species Enhancement Plan for the Chehalis River Basin [43], is the following:

$$Temp\_multiplier_{chinook\_outmigration} = \begin{cases} 1 & if\ T < 18°C \\ 1 - 0.17 * (T - 18) & if\ 18°C \leq T < 24°C \\ 0 & if\ T \geq 24°C \end{cases}$$

where $T$ is the June 1–21 ADM and $Temp\_multiplier_{chinook\_outmigration}$ is the temperature multiplier. This function was based off of the habitat suitability index for spring-run Chinook salmon developed in Raleigh et al. [60], and the temperature-survival function for spring-run Chinook salmon developed by McHugh et al. [61].

The functional relationship for spring-run Chinook salmon prespawn holding productivity and temperature (Fig 3) was developed from data in the Willamette River Basin and uses the 7-DADM as the input temperature metric [62]. While Bowerman et al. [62] included hatchery origin fish in their study, we excluded these fish when adapting the following relationship, as our analysis examines natural-origin spawners only:

$$Temp\_multiplier_{spring\_chinook\_prespawn} = 1 - \frac{e^{(-9.053 + 0.387T + 0.7919T)}}{1 + e^{(-9.053 + .387T + 0.7919T)}}$$

Here, $T$ is the 7-DADM stream temperature and $Temp\_multiplier_{spring\_chinook\_prespawn}$ is the multiplier on productivity for spring-run Chinook salmon during the prespawn holding period.

## Results

### Temperature change

In our analysis we considered stream temperatures >24˚C 7-DADM (lower limit of UILT for juvenile salmonids) to be generally unsuitable for most populations, whereas we considered

**Table 5. Percent of reaches within the Chehalis River Basin that fall under one of 3 temperature categories (< 18˚, 18˚–24˚, > 24˚) under each scenario.**

| Scenario | < 18˚C | | 18˚–24˚C | | > 24˚C | |
|---|---|---|---|---|---|---|
| Natural Potential | 33% | | 66% | | 1% | |
| Current | 17% | | 78% | | 5% | |
| | Mid-Century | Late-century | Mid-century | Late-century | Mid-century | Late-century |
| No action | 10% | 3% | 81% | 70% | 9% | 28% |
| Riparian | 14% | 5% | 79% | 75% | 6% | 21% |
| Floodplain | 11% | 3% | 82% | 73% | 7% | 24% |
| Combined | 16% | 5% | 80% | 78% | 4% | 17% |

temperatures <18˚ to be generally suitable. The proportion of reaches exceeding 24˚ increased from that of the current scenario in all future climate change and restoration scenarios except for the mid-century combined restoration scenario (Table 5). Reaches exceeding 24˚C under the no-action climate change scenario were concentrated in the upper basin near the cities of Centralia and Chehalis, as well as the mainstem Chehalis, Wynoochee, Humptulips, and Satsop Rivers (Fig 4). On the other hand, the proportion of reaches below the 18˚C threshold decreased from that of the current scenario in all future climate change and restoration scenarios (Table 5).

Despite significant potential temperature reductions (up to 6˚C or more) that can be achieved through our restoration scenarios (Fig 5), the models still projected a net increase in temperature by mid- and late-century over much of the basin (Fig 6). This occurred because much of the small stream length is in forest lands with narrow floodplains and relatively mature riparian forests, where there is little or no potential for temperature reduction (Fig 5). The proportion of stream length with potential temperature reductions >2˚C is <25%.

The magnitude of modeled temperature reduction from riparian restoration ranged from 0˚C to 6.3˚C in both the mid- and late-century scenarios, with the majority of reaches experiencing a decrease of 1.5˚C or less (S4 Fig). Reaches experiencing larger temperature reductions were located primarily in smaller tributaries with little current shading, and in agricultural or developed regions of the upper basin. In general, reaches located in forest lands of the Olympic Mountains, Willapa Hills and Cascade Foothills experienced little thermal benefit from riparian restoration because the majority of these forests are near their maximum potential tree height. However, there were some pockets of reaches with high potential for temperature reduction distributed throughout the basin.

Modeled temperatures in the floodplain scenario were generally hotter than those in the riparian scenario (Table 5). The benefits of floodplain reconnection were primarily concentrated within large river reaches, which generally have the warmest summer temperatures. The magnitude of temperature reduction from floodplain reconnection ranged from 0.29˚C in streams with < 10 m bankfull width to 1.43˚C in reaches with > 30 m bankfull width (Table 3) (S5 Fig).

The combined scenario produced the greatest reduction in modeled stream temperatures. The maximum change in temperature (-6.8˚C mid-century, -6.9˚C late-century) from the no-action scenario was higher than that of either the riparian or floodplain scenario alone, suggesting that both riparian restoration and floodplain reconnection were responsible for this reduction in temperature (S6 Fig). Still, despite the combined scenario mitigating some of the impacts of climate change, stream temperatures in this scenario were generally higher than those under current conditions.

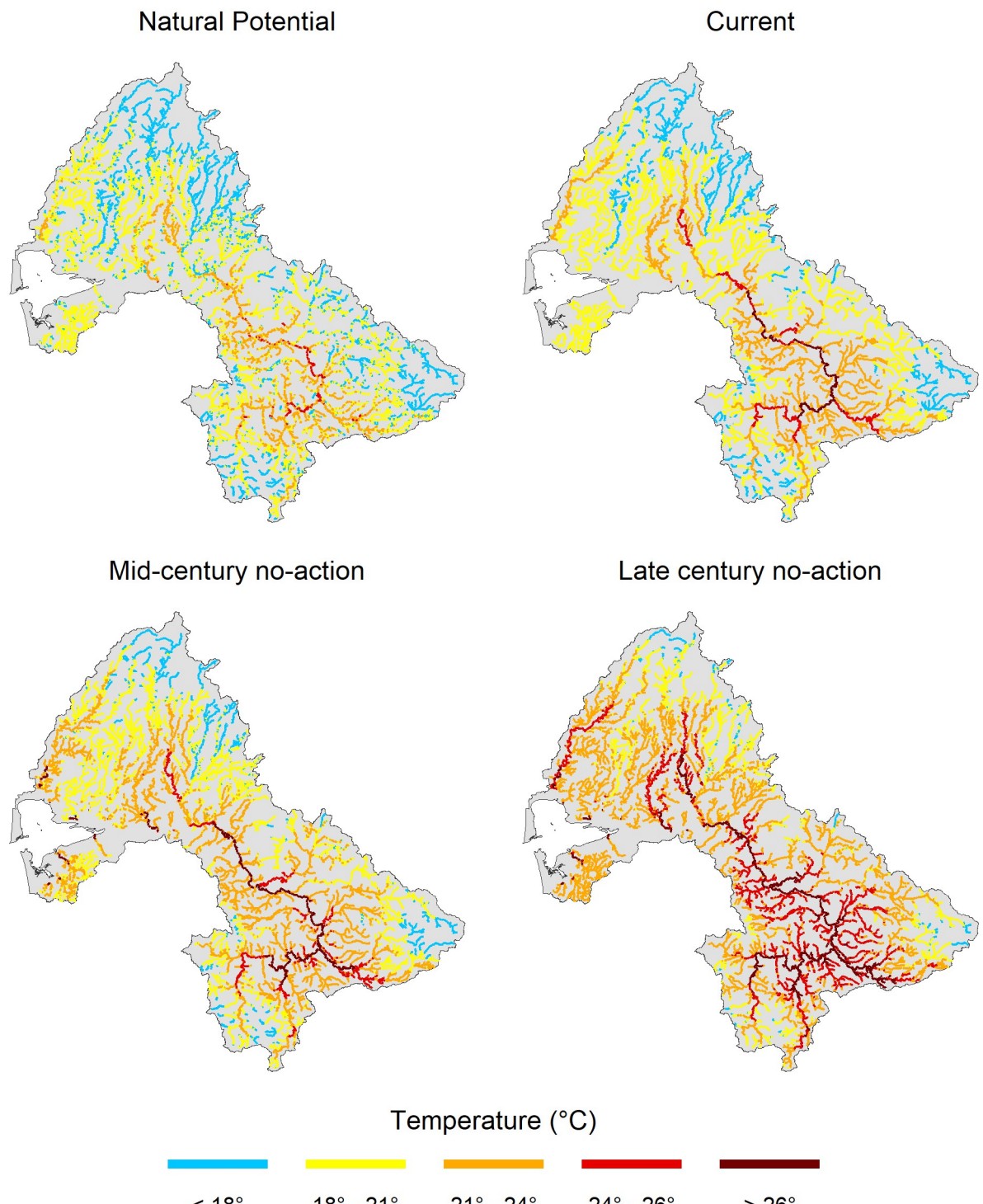

**Fig 4. Predicted August maximum seven day average of daily maximum stream temperatures (7-DADM, ˚C) under natural potential (mature tree heights, no climate change) conditions, as well as the current, mid- and late-century no-action scenarios.**

## Riparian restoration

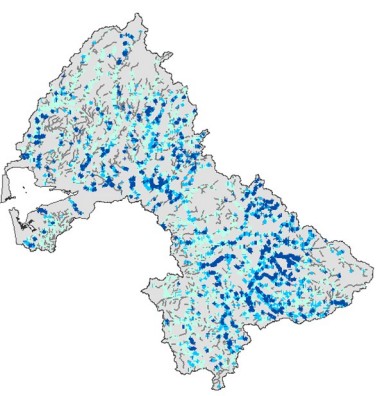

## Floodplain restoration

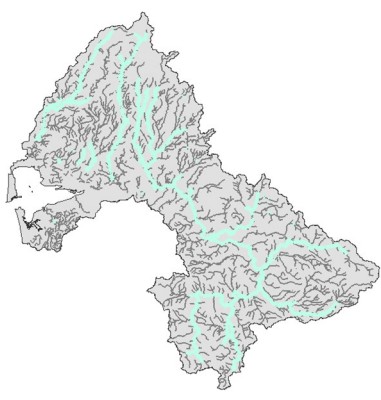

## Combined

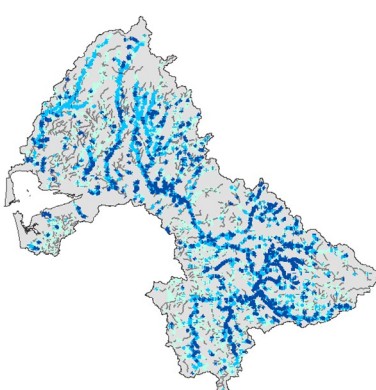

## Reduction in Temperature (°C)

> 2.5°     2.5° — 1.5°   1.5° — 0.5°   0.5° — 0°

**Fig 5. Total potential reduction in August maximum seven day average of daily maximum stream temperature (7-DADM, ˚C) due to riparian restoration (top), floodplain reconnection (middle), and a combination of riparian restoration and floodplain reconnection (bottom) by late-century.** Temperature increase due to climate change is not included in this figure.

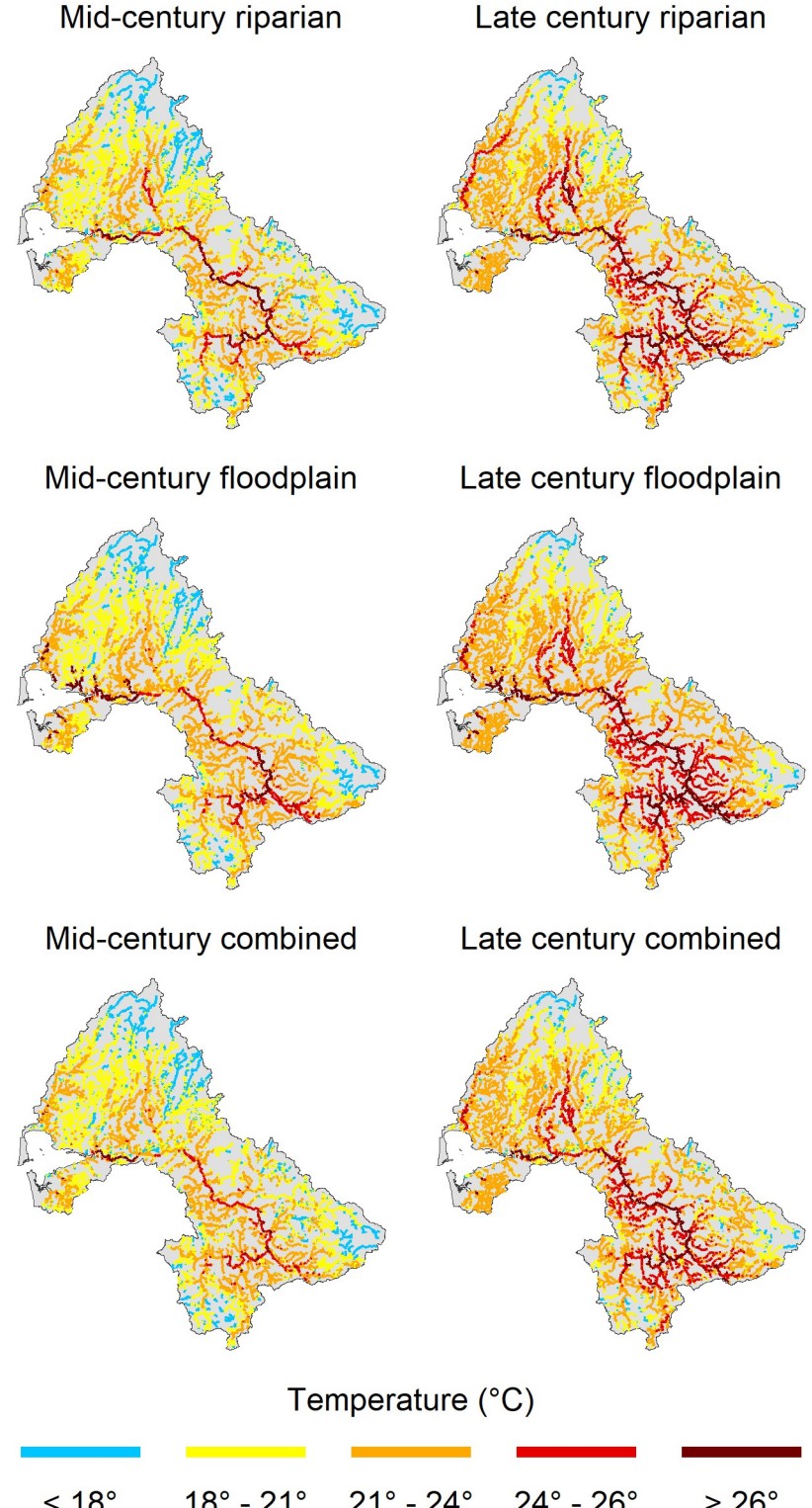

**Fig 6. Predicted August maximum seven day average of daily maximum temperatures (7-DADM, ˚C) under the riparian, floodplain, and combined scenarios for both mid-century (left) and late-century (right).**

## Spawner abundance change

The magnitude of the benefit provided by restoration differed between populations, as did the ranking of the scenarios in terms of their effect on spawner abundance (Fig 7). The combined scenario provided the greatest positive benefit for all four populations, whereas the no-action scenario had the greatest negative impact (Fig 7). The floodplain scenario provided a greater benefit than the riparian scenario for steelhead and both Chinook salmon populations for both mid-century and late-century. By contrast, the riparian scenario provided a greater benefit than the floodplain scenario for coho across both time periods.

The no-action scenario results indicated that spring-run Chinook salmon are the most sensitive population to projected temperature changes. Modeled declines in spawner abundance for spring-run Chinook salmon were 27–31% greater than for the other three populations by mid-century, and 61–78% greater than for the other three populations by late-century. Coho salmon and steelhead were similar in their sensitivity to temperature change, whereas fall-run Chinook salmon were the least sensitive to temperature change (Fig 7).

The riparian scenario benefited spring-run Chinook salmon the most, followed by coho, steelhead, and fall-run Chinook salmon. Increases in spawner abundance due to riparian restoration were less than climate change-induced decreases for all four populations, and for both mid-century and late-century, therefore, modeled net total abundance changes from the current scenario remained negative (Fig 7). However, modeled abundance change varied considerably among subpopulations. The increase in late-century spawner abundance relative to the no-action scenario resulting from riparian restoration was greatest in the Cascade Mountains Ecological Region for both coho and spring-run Chinook salmon, the Mainstem: Lower Chehalis Ecological Region for fall-run Chinook salmon, and the Olympic Mountains Ecological Region for steelhead (Fig 8).

The floodplain scenario provided the greatest percent benefit to spring-run Chinook salmon, followed by steelhead, fall-run Chinook salmon, and finally, coho. Net spawner abundance change from the current scenario was slightly positive (+1%) for steelhead for mid-century, but was negative for all other populations and time periods (Fig 7). The potential for floodplain restoration to increase spawner abundance by late-century was greatest in the Olympic Mountains Ecological Region for coho and steelhead, whereas it was greatest in the Cascade Mountains Ecological Region for spring-run Chinook salmon and the Mainstem: Lower Chehalis Ecological Region for fall-run Chinook salmon (Fig 8).

The combined scenario provided the greatest benefit of any scenario across all populations and time periods. Spring-run Chinook salmon received the greatest percent benefit of any population, followed by steelhead, fall-run Chinook salmon, and coho. Net abundance change from current for all four populations was positive for mid-century and negative for late-century (Fig 7). Late century spawner abundance increased the most in the Olympic Mountains Ecological Region for both coho and steelhead, however coho spawner abundance also increased significantly in the Cascade Mountains. Spawner abundance increased by the most in the Cascade Mountains Ecological Region for spring-run Chinook salmon, and the Mainstem: Lower Chehalis Ecological Region for fall-run Chinook salmon (Fig 8).

## Discussion

The goals of our study were to highlight differences in vulnerabilities of Chehalis River Basin salmon and steelhead populations to increased stream temperature, and to estimate the potential benefits of restoration actions targeting reduction of future stream temperatures in freshwater habitat. Our results show that significant restoration potential exists in some areas of the Chehalis River Basin that could help mitigate future temperature increases due to climate change.

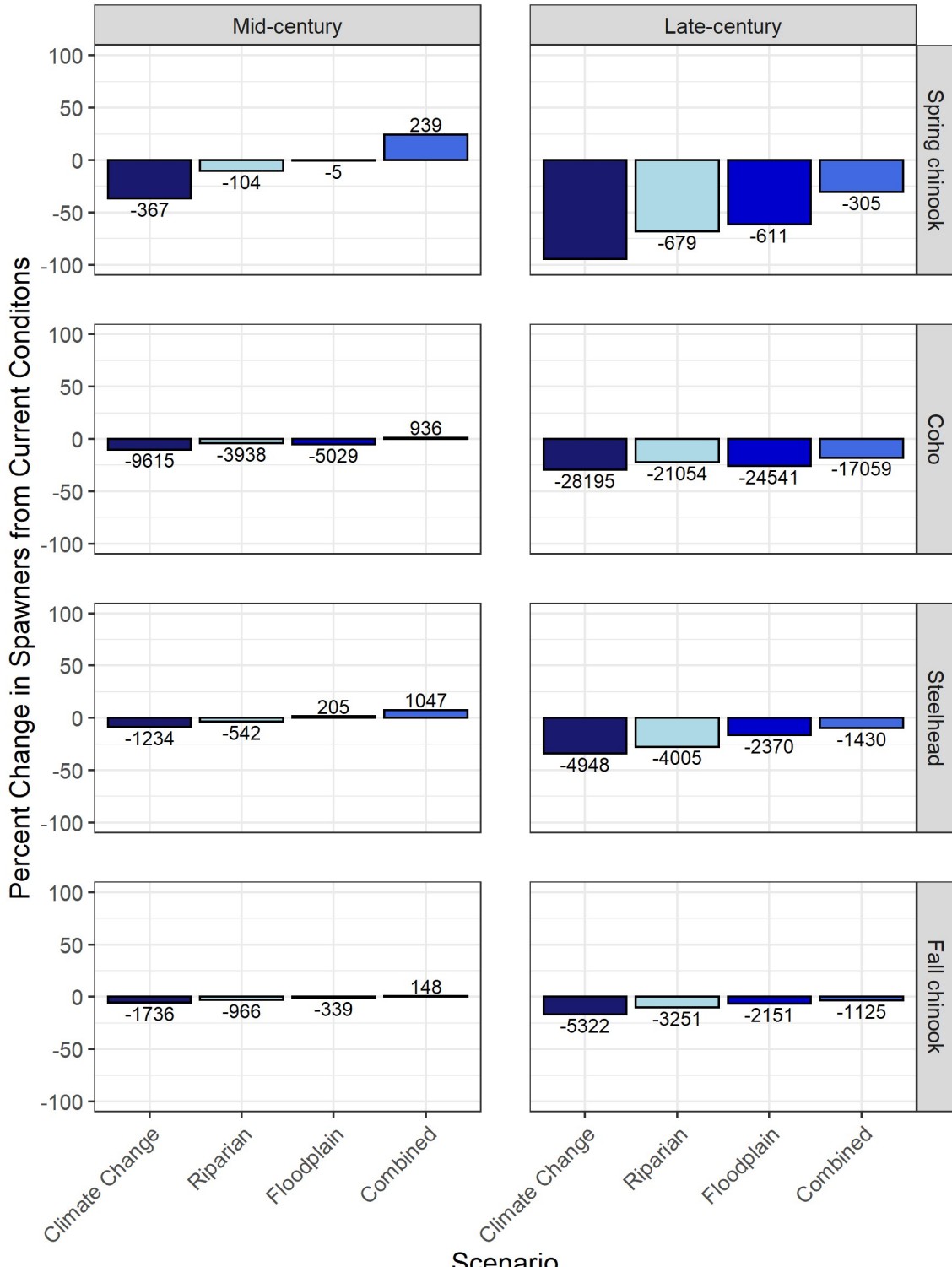

**Fig 7. Modeled percent differences in spawner abundance between current conditions and conditions under the four future temperature scenarios for all four salmon and steelhead populations.** Numbers above and below bars represent the total change in spawner abundance. Modeled spawner abundance under current conditions was approximately 1,000 for spring-run Chinook salmon, 96,000 for coho, 32,000 for fall-run Chinook salmon, and 15,000 for steelhead.

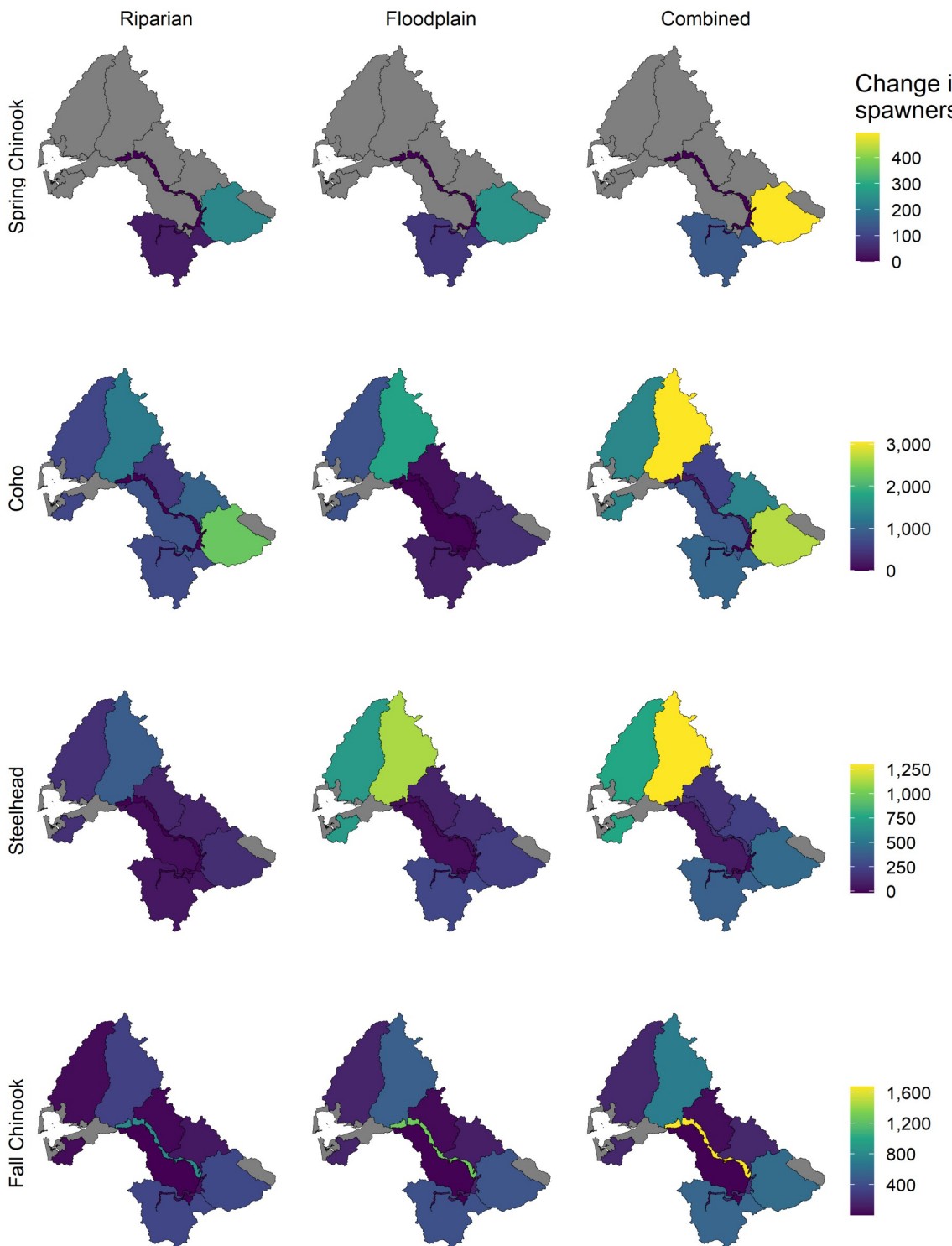

**Fig 8. Change in spawner abundance by Ecological Region for each population and restoration scenario for late-century.** Change in spawner abundance represents a change from the late-century No action scenario. Gray regions are outside of the spawning and rearing distributions for a given population.

Elevated stream temperatures pose a serious threat to spring-run Chinook salmon in the Chehalis River Basin even under our most intensive restoration scenario. Under each scenario, spring-run Chinook salmon were simultaneously the most vulnerable population to climate change, and the most responsive to modeled restoration actions. These results were similar to other studies in which spring-run Chinook salmon populations were particularly vulnerable to increased stream temperatures [13, 64]. Coho, steelhead, and to a lesser extent, fall-run Chinook salmon are also likely to undergo population declines due to increased stream temperature, but may also benefit from restoration targeting increased stream temperatures. A combination of riparian restoration and floodplain reconnection would be the most effective approach in mitigating temperature increases within the basin for all four populations. However, riparian restoration would likely be the most effective single action for coho salmon, whereas floodplain restoration may be the most effective single action in managing the other three populations.

An important outcome of our modeling is that floodplain reconnection is likely to be more effective at reducing temperatures in large rivers, whereas riparian restoration is likely more effective at reducing temperatures in small streams. Temperature reduction via hyporheic exchange is a function of flow path length [29, 57, 58], which tends to increase with increasing river and floodplain size [30]. Therefore, our model projects that larger rivers and floodplains have relatively large temperature reductions when floodplains are reconnected, whereas small streams have less benefit from floodplain reconnection and many small streams have no floodplain reconnection potential at all [35]. By contrast, riparian shade is more effective at reducing temperature in small streams because relatively small trees can shade narrow channels, whereas even tall trees can only partly shade larger rivers [25]. Hence, our model projects the greatest temperature reduction potential on small, unshaded streams where planting trees can increase shade within a few decades. We did not model effects of beaver dams or sediment retention by wood as a potential mechanism that may also decrease temperature [65, 66], but both actions would likely be more effective in small streams.

Restoration actions and projected stream temperature increases were predicted to impact populations differently depending on freshwater habitat use and life history characteristics [36]. Our results suggest that restoration of riparian shading would most benefit populations rearing in small streams over the summer. Coho salmon occupy small streams over summer at higher densities than any of the other three populations [36]. Additionally, the rearing distribution for juvenile coho in the Chehalis River Basin is more extensive within small stream habitat than that of any of the other populations (S3 Fig). Accordingly, coho salmon benefited more from restoration of riparian shading than from floodplain restoration in our model.

On the other hand, floodplain restoration is likely to benefit populations rearing and outmigrating in large rivers over summer. Juvenile steelhead and Chinook salmon are more concentrated in large rivers over summer [36], and the spring-run Chinook salmon rearing distribution is almost entirely within large river habitat (S3 Fig). Consequently, floodplain restoration provided a greater positive impact than restoration of the riparian canopy for steelhead, and both Chinook salmon populations. Our analysis suggests that floodplain restoration alone could potentially mitigate most of the impact of mid-century temperature increases on spring-run Chinook salmon in the Chehalis River Basin, but would not be enough to offset late-century temperature increases.

A combined approach of restoring both riparian shading and floodplain connectivity would provide the greatest benefit to all four populations. This was especially true for the spring-run Chinook salmon population, which experienced a net gain in spawner abundance of 24% by mid-century despite increases in temperature from climate change. The combined scenario is particularly effective due to its benefits to populations that rear and outmigrate over summer in small streams and large rivers alike.

## Model uncertainties

Stream temperature inputs are a key source of parameter uncertainty in our life-cycle model. Sources of uncertainty in the stream temperature projections include air temperature and stream flow projections, as well as shade and floodplain connectivity effects. Sources of uncertainty in air temperature projections include uncertainty in both emissions scenarios and climate models [17, 52], which combined to produce projected air temperature increases ranging from approximately +2˚C to +6˚C in the 10 GCM ensemble used in the IPCC A1B scenario [67]. Projected decreases in summer stream flow are also likely to exacerbate future warming within rivers [6, 67]. A separate modeling effort for future stream flows in the Chehalis Basin showed that estimates of future changes in low flows are relatively similar among climate scenarios (RCP 4.5 and 8.5), but can vary widely between hydrologic models and between snowmelt-dominated and rainfall-dominated hydrologic regimes [68].

The uncertainties in future air temperature and stream flow illustrate that the temperature change projections used in our model might over- or under-estimate future temperatures to an unknown degree, although we cannot quantify this uncertainty without additional stream flow and temperature modeling efforts. There is also a possibility that the conversion equations used to translate August ADA into June 1–21 ADM (S2 Fig) might not capture changes to this correlation that might occur in the future. For example, if June temperatures were to increase as much as the August Temperatures in the model, the decline in fall-run Chinook salmon spawner abundance would be twice as large as our current result, and the percent decline would be similar to those of coho and steelhead. The spring-run Chinook salmon population would near extinction in all scenarios except for the combined riparian and floodplain restoration scenario, however the conclusion that spring-run Chinook salmon are the population most-vulnerable to stream temperature increase driven by their vulnerability during pre-spawning would be unchanged. Thermal tolerances of species were represented by simplified functional relationships to temperatures in our model, and uncertainty around those relationships is not accounted for in the projections. Several of the thermal tolerance curves used in our model were developed in a laboratory setting, and those developed within rivers came from outside of the Chehalis River Basin. Salmon species can also potentially adapt to the effects of climate change to some degree [69], and plasticity in timing of emergence and juvenile growth may reduce climate change effects. For example, increased stream temperatures may potentially lead to an earlier start to Chinook salmon growth, and thus an earlier start to outmigration [70]. In our model, a shift to an earlier outmigration period could allow a greater proportion of both Chinook salmon populations to avoid the negative impacts of high summer stream temperatures.

The reach-scale temperature modeling described in this paper fails to capture sub-reach-level heterogeneity in stream temperature (i.e. thermal refuges) that could be exploited by fish during the hottest periods of the day or year. Studies have shown that salmonids move between colder and warmer patches of rivers over short time periods in order to take advantage of different thermal regimes [44, 71–73]. Given spatial differences in the rearing distributions of the four populations (S3 Fig), potential thermal refuges in the basin will disproportionately benefit certain populations.

The ecological effects of warmer stream temperatures on salmonids will also likely vary with food availability [15]. Because food availability modifies the effects of water temperature on salmon growth rates (e.g., [74]), we simulated the effect of changing food availability by adjusting the lower end of the temperature survival curve for coho, to reflect an increased or decreased tolerance of high temperatures based on food availability [15]. We found that an increase in thermal tolerance of 1–3˚C could boost coho spawner abundance in our

temperature scenarios by as much as 7–31%. On the other hand, a reduction in thermal tolerance of 1–3˚C could decrease coho spawner abundance in our temperature scenarios by as much as 8–23%. Therefore, actions that target increasing food sources for salmon are likely to be a highly beneficial addition to restoration plans in the Chehalis River Basin. There is little research on restoration actions that might increase food availability, but wood augmentation, floodplain reconnection, creating light gaps in riparian forests, and altering riparian vegetation species compositions can increase prey [75–80].

## Other considerations

Projected impacts of climate change on freshwater ecosystems, including elevated stream temperatures [5, 6, 17, 19], and changes to freshwater flow regimes [5, 6, 12, 19] highlight the importance of climate refuges in sustaining future salmonid populations [81]. While other studies suggest that changing ocean conditions due to climate change could play a significant role in limiting Pacific salmon populations [82, 83], the goal of our study was to assess the potential for mitigation of temperature increases in fresh water, and therefore we did not model changes to spawner abundance under differing ocean conditions. However, restoration planning for salmonid recovery and future resilience could include other aspects of climate change, including impacts of changing ocean conditions. Additionally, in the Chehalis River, non-native species, including smallmouth and largemouth bass are prevalent throughout the main stem, major tributaries, and floodplain habitat, and likely negatively affect salmon population through various pathways, including direct predation [84]. These effects are anticipated to increase under climate change scenarios, although restoration might limit them somewhat [85]. Our model does not account for potential non-native species impacts, and therefore actual climate change effects on native salmonids could be greater than those presented in our results.

## Conclusions

Our results highlight differences in both the vulnerability of four salmonid populations to future increases in stream temperature, and the effectiveness of three restoration scenarios in mitigating these temperature increases. While fall-run Chinook salmon were relatively resilient to increasing stream temperatures, our analysis indicated that coho, steelhead, and especially spring-run Chinook salmon in the Chehalis River Basin may undergo major declines by the end of the century without management intervention to mitigate the thermal impacts of climate change. Our model suggests that restoration opportunities exist within the Chehalis River Basin that could mitigate climate change-induced declines for all four populations, and that sustaining vulnerable salmonid populations in the Chehalis River Basin into the future will likely require a management approach that prioritizes limiting increases in stream temperature.

While our model illustrates the potential benefit of restoration actions that target thermal regimes, it is also important to consider other habitat factors, restoration actions, and climate change impacts when developing a restoration plan. Even under our best-case temperature restoration scenario, late-century spawner abundance declined in all four populations due to increased stream temperature. This suggests that restoration actions targeting temperature alone will not likely be sufficient to increase salmon populations over the long term. Furthermore, studies have shown that other restoration actions such as recolonization by beavers or placement of beaver dam analogs [86, 87], removal of barriers to fish passage [88], and placement of large wood [89, 90] can successfully increase salmonid populations by addressing other causes of habitat degradation. While we do not explore these other restoration actions in

this paper, our model framework does support evaluations for other restoration actions including those listed above, and recent papers by Beechie et al. [35] and Jorgensen et al. [36] highlight other restoration opportunities within the Chehalis River Basin that may help sustain salmon populations in a future climate.

## Supporting information

**S1 Fig. Ecological Regions (colored regions) and subbasins of the Chehalis River Basin.** Gray regions are not included in the Ecological Regions.
(TIF)

**S2 Fig. Conversion between August Average Daily Average (ADA) and August 7-Day Average Daily Maximum (7-DADM) (left), and June 1–21 Average Daily Maximum (ADM) (right).** Observed temperature data were used to calculate August ADA (n = 80 sites), 7-DADM (n = 80 sites), and June 1–21 ADM (n = 43 sites). Blue line represents the line of best fit, and grey shading represents the 95% confidence interval.
(TIFF)

**S3 Fig. Spawning and rearing distributions for spring Chinook salmon, fall Chinook salmon, coho salmon, and steelhead in the Chehalis River Basin.**
(TIFF)

**S4 Fig. Temperature reduction resulting from riparian restoration for mid-century (left) and late-century (right).**
(TIFF)

**S5 Fig. Temperature reduction resulting from floodplain reconnection for both.** Temperature reduction is the same for both mid-century and late-century.
(TIFF)

**S6 Fig. Temperature reduction resulting from a combination of riparian restoration and floodplain reconnection for both mid-century (left) and late-century (right).**
(TIFF)

## Acknowledgments

We greatly appreciate the many people whose contributions have made this project possible. Collaboration, inputs, and model reviews were provided by Jamie Thompson, Spencer Kubo, John Ferguson, Neala Kendall, Larry Lestelle, and Gary Morishima. Helpful reviews of the manuscript were provided by Aimee Fullerton, Lisa Crozier, and Joe Anderson, Jon Honea, and one anonymous reviewer.

## Author Contributions

**Conceptualization:** Caleb B. Fogel, Colin L. Nicol, Jeffrey C. Jorgensen, Timothy J. Beechie, Peter Kiffney, Gustav Seixas, John Winkowski.

**Data curation:** Caleb B. Fogel, Colin L. Nicol, Jeffrey C. Jorgensen, Timothy J. Beechie, Britta Timpane-Padgham, John Winkowski.

**Formal analysis:** Caleb B. Fogel, Colin L. Nicol, Jeffrey C. Jorgensen.

**Funding acquisition:** Timothy J. Beechie.

**Investigation:** Caleb B. Fogel, Colin L. Nicol, Jeffrey C. Jorgensen, Timothy J. Beechie, Britta Timpane-Padgham, Peter Kiffney, John Winkowski.

**Methodology:** Caleb B. Fogel, Colin L. Nicol, Jeffrey C. Jorgensen, Timothy J. Beechie, Gustav Seixas.

**Project administration:** Timothy J. Beechie.

**Resources:** Caleb B. Fogel, Colin L. Nicol, Jeffrey C. Jorgensen, Timothy J. Beechie, Britta Timpane-Padgham, Gustav Seixas, John Winkowski.

**Software:** Caleb B. Fogel, Colin L. Nicol, Jeffrey C. Jorgensen, Britta Timpane-Padgham, Gustav Seixas.

**Supervision:** Timothy J. Beechie.

**Validation:** Caleb B. Fogel, Colin L. Nicol, Jeffrey C. Jorgensen.

**Visualization:** Caleb B. Fogel, Colin L. Nicol, Jeffrey C. Jorgensen.

**Writing – original draft:** Caleb B. Fogel.

**Writing – review & editing:** Caleb B. Fogel, Colin L. Nicol, Jeffrey C. Jorgensen, Timothy J. Beechie, Britta Timpane-Padgham, Peter Kiffney, Gustav Seixas, John Winkowski.

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
