## [Decision Letter · Decision Letter 0]

9 Dec 2021

PONE-D-21-16705How riparian and floodplain restoration modify the effects of increasing temperature on adult salmon spawner abundance in the Chehalis, River, WAPLOS ONE

Dear Dr. Fogel,

Thank you for submitting your manuscript to PLOS ONE. After careful consideration, we feel that it has merit but does not fully meet PLOS ONE’s publication criteria as it currently stands. Therefore, we invite you to submit a revised version of the manuscript that addresses the points raised during the review process.

Please carefully address the thorough and constructive comments provided by both reviewers. In particular, they both identify concerns with clarity in methodology; several unpublished papers are referenced, and incomplete information is provided in the methods section of this paper. Please provide updated references (if available) and additionally provide further detail on methodology within this manuscript.  Additionally, I would like to see further clarity on how this manuscript is clearly distinguished from other, related, manuscripts that were under review at the time of submission. With these updates and careful attention to reviewer comments, I would be pleased to see a revised version of this manuscript. ==============================

We look forward to receiving your revised manuscript.

Kind regards,

Rachel A Hovel

Academic Editor

PLOS ONE

Journal Requirements:

2. In your Methods section, please provide additional location information of the study area, including geographic coordinates for the data set if available.

4. We note that Figures 1, 2 , 5, 6, 7 and 9 in your submission contain map images which may be copyrighted. All PLOS content is published under the Creative Commons Attribution License (CC BY 4.0), which means that the manuscript, images, and Supporting Information files will be freely available online, and any third party is permitted to access, download, copy, distribute, and use these materials in any way, even commercially, with proper attribution. For these reasons, we cannot publish previously copyrighted maps or satellite images created using proprietary data, such as Google software (Google Maps, Street View, and Earth). For more information, see our copyright guidelines: http://journals.plos.org/plosone/s/licenses-and-copyright.

a. You may seek permission from the original copyright holder of Figures 1, 2 , 5, 6, 7 and 9 to publish the content specifically under the CC BY 4.0 license.  

Reviewers' comments:

Reviewer's Responses to Questions

**Comments to the Author**

1. Is the manuscript technically sound, and do the data support the conclusions?

Reviewer #1: Partly

Reviewer #2: Yes

2. Has the statistical analysis been performed appropriately and rigorously? 

Reviewer #1: I Don't Know

Reviewer #2: N/A

3. Have the authors made all data underlying the findings in their manuscript fully available?

Reviewer #1: No

Reviewer #2: No

4. Is the manuscript presented in an intelligible fashion and written in standard English?

Reviewer #1: Yes

Reviewer #2: Yes

5. Review Comments to the Author

Reviewer #1: I reviewed the Research Article manuscript PONE-D-21-16705 entitled “How riparian and floodplain restoration modify the effects of increasing temperature on adult salmon spawner abundance in the Chehalis, River, WA”. This study quantified how much two restoration actions, floodplain restoration and riparian restoration, would reduce stream temperature relative to no-action under climate change scenarios. They then evaluated future habitat conditions and salmonid capacity in the Chehalis River basin for a variety of scenarios (historical thermal conditions, current thermal conditions, future thermal conditions with no-action, future thermal conditions with one restoration action, future thermal conditions with both restoration actions). They find that implementing both restoration actions would result in the greatest decrease in stream temperature, and hence the greatest benefit for salmonids, although specific populations and life stages might benefit more from one restoration action.

The amount of work that the authors did is commendable, and this is an important study on how restoration can mitigate negative impacts of climate change for declining and threatened salmonids. However, I cannot currently recommend that this manuscript be accepted into PLOS ONE without major revisions. Major problems are present in the Methods and analyses. The authors sometimes based their analyses from other sources with citations (e.g. the HARP and SHaRP frameworks from Beechie et al. 2021 and Jorgensen et al. in review); however, they need to describe these previous results, especially for works that are not publicly available. I was unclear on how they came up with some of their thermal conversion equations, scaling parameters, and other equations. I also think I found some potential errors in equations as written (e.g. line 326). Background for these needs to be more thorough, or equations need re-analyzed. Although unclear, it seems that this manuscript combines the frameworks from Beechie et al. (2021) and Jorgensen et al. in review to assess salmonid capacity under different restoration scenarios. It might be beneficial to 1) publish this as a companion paper, or 2) borrow or adapt a figure from the previous two papers to help illustrate the framework (including equations and parameters borrowed from those papers) more clearly.

Most other problems are with the writing. The manuscript is too long, is sometimes redundant, is sometimes unclear, and needs some reorganization. There are also too many figures, some of which could be combined, relegated to supplement, or removed. I provide detailed comments below to help the authors improve their manuscript for future publication. As stated earlier, I think this will be an important contribution showing how to mitigate negative climate change impacts on salmonids. I would be more than happy to review a revised version.

ABSTRACT

General thoughts: It was hard to tell when the authors transitioned from Intro to Methods to Results. It might be more useful to readers to state number of parameters, type of model, and general conclusions in the Abstract.

Line 21: maybe substitute “cold-water” for “anadromous” to point out that an increase in stream temperature is not good for salmonids because they require cold water. You state that your study is on “anadromous salmonid populations” in line 24 – no need to point this out twice in the abstract.

Line 29: “We applied a temperature effect” is vague. Do you mean an temperature effect term in your life cycle or habitat model?

Line 31-32: remove “…and therefore experience the warmest summer temperatures,”

Line 30-32: Unclear here why it is important to note that steelhead are more tolerant of higher temperatures than coho. Is this a result, method, or observation?

Line 38-41: Great sentence for an Abstract. Add ‘%’ to -34

INTRODUCTION

Line 46: Change “increasing” to “warming”

Line 47: Change “increases in sea level” to “rising sea levels”

Line 48: Awkward. Sounds like occurrence and intensity of ocean acidity will increase. Rephrase.

Line 52: Remove or relocate “of cultural, economic, and ecological significance”

Line 52: Add final sentence here stating something like, “For species already negatively impacted by human activities [Or, for threatened or declining freshwater species], it is imperative to assess how climate change may impact future habitat quality so that…”. Then, start new paragraph after.

Line 56-59: Awkward.

Line 59-61: Don’t discount that warming temperatures could also be beneficial to some life stages, such as juveniles, which grow faster at warm temperatures (to a point) and could exploit warmer habitats. E.g. Armstrong et al. 2021 Nature Climate Change https://www.nature.com/articles/s41558-021-00994-y

Line 69-71: I think this sentence is the main point of this paragraph – that few salmonid climate change studies quantify how restoration could mitigate negative impacts to populations. Move this idea to the first sentence. May need to rearrange thoughts for flow.

Line 72-76: Move to preceding paragraph.

METHODS

General thoughts: Change the “Study Area/Overview” section to “Study Area and Species”. Shorten this section to description only. Add a new section called “Overview” or “Approach” or similar, where you give a short overview of your approach. This will help the reader to follow your methods logically as you talk about each dataset.

Line 91-93: Awkward. Maybe, “The Chehalis River Basin consists of 63 subbasins [33,34] and 10 Ecological Regions as defined by…”

Line 103: Need citation.

Line 135: Simplify “through which we explored” to “to explore”

Line 137: “Each future temperature scenario was modeled…”

Line 137: late-century

Line 142-145: Awkward.

Line 146: Change to “Based on reach-level projected stream temperature for each scenario, we estimated…”

Line 148-149: You have to give a brief description of the SHaRP model framework here, especially since Jorgensen et al. is still under review.

Line 149: “the August 7-DADM” is unclear. Do you mean the 7 day period with the warmest 7-DADM in August?

Line 150-152: Simplify to “Our modeled restoration scenarios assumed that…”

Line 157: It might be more readable to divide this into two sections: “Current temperatures” and “Future Temperature Change—No-action”. ----

Line 158-163: Add a brief sentence or two describing the Chehalis Thermalscape spatial stream network model, and its results, since the report is under review and not currently available to readers.

Line 163-164: Why did you segment a stream into ~200m-long reaches? It seems like it might be better to use 500-1000m-long reaches, to match the stream temperature dataset. Need to address this.

Line 167: “different” from each other, or different from the stream temp dataset metrics?

Line 167-176: This paragraph is confusing. I understand why the authors need the conversion equations, but I’m unclear why they chose to look at a different temperature dataset with 80 sites of daily stream temp (WDFW Riverscape) to calculate the conversion equations. Were you unable to access the Chehalis Thermalscape project original stream temperature data (presumably, daily temps)? Need a sentence to explain your rationale, or need to calculate conversion equations from the Chehalis Thermalscape project data.

Line 171: need citations and explanations of why each population/life stage function is different.

Line 184: need citations and short descriptions of the A1B emissions scenario and the VIC model.

Line 187: I thought the Chehalis Thermalscape predicted ADA temps, not 7-DADM. Address.

Line 191-195: This is precisely how all previous models should be mentioned, with a short description of their model and results, and how you applied it for your study.

Line 196: Where did you obtain or how did you calculate A, the drainage area?

Line 199: Replace “the canopy opening angle” with “”

Line 199-205: Again, briefly describe lidar, and describe how you defined height class from the lidar data or aerial imagery.

Line 208-212: Great idea! Again, briefly describe these growth rates. Are they temperature-dependent or independent? What is the relationship between tree growth, tree height, and ?

Line 216: Does the Chehalis Thermalscape include canopy opening angle in its model? Drainage area? How different are the predicted current temps between the Chehalis Thermalscape project and Seixas et al. model? You need to show that Tdiff(growth) is attributed to growth, not differences between the two temp models.

Line 231-233: Confusing statement. Do the ASRP guidelines dictate that you estimate the widths of floodplain corridors? Clarify.

Line 231 (or so): Somewhere, add a transition phrase: “To define the scaling parameter, we…”

Line 230 on: So, is the scaling factor based on a linear relationship of 0-425 m width and 0-2°C? Is this based on the authors’ assumed estimate or from the study cited above? I am concerned that the authors are defining a linear relationship that may not be linear. Please elaborate, clarify, or revise.

Line 235-236: Does it matter to distinguish between large and small river reaches?

Line 242: Use subscripts to make this equation easier to read. Also, need to elaborate on this relationship since it is from another reference.

Line 249: Need to briefly describe the SHaRP for your readers. Is that what you do following? Unclear.

Line 251-252: Unclear after “is affected through…” Revise.

Line 261-266: When borrowing estimates from other sources, briefly describe their work so that readers know a bit about those estimates. For example, “[33] combined geospatial layers, recent field surveys, and models to estimate historical and current habitat conditions in the Chehalis River basin.” Elaborate on your capacity and productivity multipliers.

Line 279: Remove “based on the 7-DADM stream temperature”

Line 277 on: Combine the temperature multipliers sections into a single section for clarity. Give a short introduction as to what the temperature multiplier is for, then talk about your multiplier for each life stage/population in separate paragraphs. Once you do this, you can reduce a lot of redundant text.

Line 308-311: Rephrase to something like, “Although juvenile Chinook salmon can outmigrate from xx through xx, 45% of parr oumigrate between June 1-21 [45].” This keeps the focus that you are still estimating thermal habitat during the peak of the outmigration.

Line 326: Check this equation. First, it is an odd way to write this, making me think there is an error. Second, I cannot find this equation in ref 55. Third, plugging in temperatures from 12-24 C makes the temp_multiplier a maximum of 0.006. Is that correct??

RESULTS

Line 331: Here or somewhere in the Methods, add a sentence to explain why you are evaluating these threshold categories. This could be something like: “To be protective of all salmonid life stages and populations, we defined temperatures above 24C as unsuitable.” Or, “Temperatures <18C are generally good for all life stages.”

The ’Temperature Change’ section is very long. Condense. From my view, your results are essentially: 1) Temperature is predicted to increase with climate change. 2) Each restoration scenario reduces the temperature relative to no action. 3) Combining restoration scenarios produced the greatest reduction in modeled stream temperature. 4) Restoration actions will still result in warmer temperatures.

Line 393-399: Nicely written.

Line 406: indicated

Line 412-442: Verbose. Condense.

DISCUSSION

I prefer to see the first sentence of the Discussion as a short summary of what you did and why. Then, lead with your most important result or conclusion. Is your most important result that salmonid populations will be impacted differently by climate change?

Line 449-451: Awkward

Line 452-465: nicely written

Line 469-470: Move the phrase after “whereas the opposite…” to ~line 475 “On the other hand…”

Line 480-493: First, juvenile outmigration here refers only to parr, correct? What about yearlings that rear over summer? Second, why the focus on this paragraph on only Chinook and not coho and steelhead as well?

Line 484-485: Maybe – but June is peak outmigration time, right? So there might be a benefit to entering the ocean then compared to earlier in the year (maybe juveniles are larger and can more easily withstand predation (e.g. Beamish and Mahken 2001), or maybe juvenile prey is more abundant then). If this is the case, a negative impact during peak timing is still damaging to the population, right?

It might be beneficial to reorganize a few paragraphs in the Discussion to avoid redundancy and to increase flow and clarity. My views are: 1) Short summary paragraph of most important results. 2) Leave second paragraph as is. 3) Keep the focus on restoration actions. “Restoration actions were predicted to impact salmonid populations differently, dependent on freshwater habitat use and life history characteristics.” Then something like, “Riparian shading would greatly benefit salmonids over-summering in small streams.” Then discuss which populations riparian shading helps and why. 4) “On the other hand, floodplain restoration was predicted to strongly benefit salmonids oumtigrating or rearing during the summer in large rivers. Juvenile steelhead and Chinook salmon…” 5) “We found that implementing both restoration actions would result in the lowest increases in stream temperature relative to no-action. Implementing both actions is particularly important for spring-run…”

Line 506-521: This can be condensed. It’s OK that you did not model every future temperature scenario. Simply state that other air temp projections could result in cooler or warmer stream temperatures than the A1B scenario used in [30].

Uncertainty in stream temp projections: Can you condense the first three paragraphs into one? They seem to be making the same point regarding uncertainty in air temp and flow projections.

Line 583: change “are” to “were here”

Line 589 (or so): Additionally, other factors can change these thermal relationships. For example, food availability, flow velocity, or predator density. Add a sentence or two here describing these impacts on thermal tolerances, and implications for your results.

Line 590-606: So, are you saying that only fall-run Chinook have the potential to change their phenology to avoid climate change? Why or why not?

Line 607-609: This seems like it should fit better in the “Uncertainty in stream temp projections” section. Maybe change section name from projections to modeling. Then elaborate. The rest of the paragraph is probably fine here.

Line 637-638: Awkward phrasing. Maybe simplify to “An increase in thermal tolerance by 1-3C could boost coho…”

Line 640: as above.

Line 633-657: Interesting ideas, to increase thermal tolerance by increasing prey availability, and thereby help salmonids cope with climate change. I think there might also be research regarding increasing prey availability on floodplains. Add a citation/sentence if you have one handy.

Recycle the “Other Considerations” sentences to other sections. Condense.

Line 684-696: I think this is a great first paragraph of your Discussion.

Good final paragraph.

Throughout your Discussion, you find that spring-run Chinook salmon are particularly vulnerable. This result is also replicated in other studies. Start with FitzGerald et al. 2021 Global Change Biology and references within. Add to your Discussion to bolster the results of your study.

REFERENCES

Ref. 33: Update

Ref 49: Not all info included.

FIGURES AND TABLES

Table 1: nice table. This is not necessary, but it may strengthen your paper if you have hypotheses for each scenario. For example, “Riparian restoration” is hypothesized to cool temperatures in the summer, which helps salmonids present in the summer (i.e. Chinook spring-run pre-spawn adults, juvenile coho, juvenile steelhead), but may not impact fall-run Chinook. I think most of this information is present in your Study Area/Overview section.

Table 4: Put spring-run and fall-run Chinook salmon on separate rows. You repeatedly talk about the four salmonid populations, so keep that cohesive. Then, you can remove the footmark.

Table 5: This table might present better as a figure. For example, a bar plot showing how % for each temp category changes from natural potential to current to mid-century to late-century, with each restoration action as different groupings.

Fig.1: nice figure. The yellow for agriculture gets lost a bit, even at high res – maybe brighten to orange, or darken the green for forest. Add ‘Columbia River’ to inset map for reference to those not familiar with the region. Is the forest/agriculture/developed characteristics important for your model, or just as interesting background?

Fig. 2: nice figure, but is this necessary for your model? Might be better as a supp fig.

Fig. 3: Change y axis to “August 7-DADM” for clarity. Add (°C) to all axes.

Fig. 4 legend: Check your reference numbers here. Are these really the best references for these thermal performance curves for each species?

Fig. 5. Very nice figure.

There might be a way to condense Fig. 5-7 into a single figure. Essentially, a multi paneled map fig with rows vs. columns as time vs. restoration actions. You could show predicted temperature or predicted temperature changes and still get your point across more simply.

Fig. 8 legend: Need citation for estimated spawner abundances.

Fig. 8: nice figure.

Fig. 9: Why do you show spawner abundance by ecological region rather than along stream reaches? It might be more helpful to show calculated values (spawners per reach), or show % of maximum spawners in entire basin for each reach. Also, what is the gray?

SUPPLEMENTARY FIGURES

These are the figures (S2-S4) I was thinking about earlier with combining Figs. 5-7 into a new simpler figure. Think about if you could combine them. If you cannot, think about if you need legends for the supp figs.

Reviewer #2: Fogel et al.’s “How riparian and floodplain restoration modify the effects of increasing temperature on adult salmon spawner abundance in the Chehalis River, WA” describes a multi-model exercise to evaluate estimated impacts of future water temperature increase and habitat restoration on 4 salmonid populations that reproduce in the Chehalis River. They find that modeled increases in water temperature due to climate change will reduce spawner numbers of all four populations at both mid-century and late century. However, modeled habitat restoration shows the potential to at least partially ameliorate those impacts. In particular, modeled restoration of riparian forests has the potential to benefit life stages occupying smaller streams and modeled restoration of floodplain connectivity has the potential to benefit life stages occupying wider channels such as the main stem.

The lack of availability of some important details of their work means I cannot fully evaluate this manuscript. At least three key sources describing their life-cycle models as well as their main stream temperature model are under review and weren’t made available to me. The life-cycle models estimating salmon spawner abundance as influenced by stream temperature are described in Beechie et al. and Jorgensen et al., citations 33 and 34, both under review at PLoS ONE. Likewise, the details of their main stream temperature model, the Chehalis Thermalscape spatial stream network temperature model, can be found in Winkowski and Zimmerman, citation 30, which is under review at the Washington Department of Fish and Wildlife.

If those works pass review without substantial changes, there remains the question of whether the models they elaborate were used appropriately here. For example, I would like to see how productivity and capacity are used to determine the number of fish surviving each life history stage. Line 289 indicates that a temperature multiplier was applied to both productivity and capacity. This is different from some life-cycle models where that would be appropriate only for productivity.

That said, if the three sources describing their key models pass review and if they are used appropriately here, then this work is an important contribution to the management and restoration of salmonid populations in the face of climate change. I see no need for the authors to run additional model scenarios. However, I do recommend below some changes to provide important clarification.

The authors indicate on line 185 that the estimates of future water temperature and stream flow produced by the water temperature–climate model they used were based on the A1B emissions scenario. I recommend adding a sentence explaining why the A1 storyline and scenario family and why the B group of the A1 family are appropriate for this exercise and what the use of A1B assumes about the global carbon emissions pathway. Comparing A1B to RCP scenarios 4.5 and 8.5 beginning at line 518 is useful, but there too an indication of what the RCP scenarios assume about carbon emissions pathways would be useful context.

186: recommend specifying “water” temperature (or “air” if that’s what they mean here)

At line 230, the authors describe using water temperature responses to riparian reconnection in a wider channel section of the Willamette River to scale responses for narrower channel sections on the Chehalis, producing Table 3. They should explain how they produced a linear scale from just one point (425, 2) or indicate that more than one width-temperature point was available from the Willamette.

This reader was confused about the mention of a multiplier at line 256 without further elaboration until line 278. I recommend adding something like “described below” to the end of the sentence on line 256.

The use of % is ambiguous on Lines 332–338, 362, 364, 375, 375, 387, 389, and 390. For example, beginning on line 331, they state that “the number…increased from the current scenario by 4%.” However, a 4% increase of 5% might mean an increase of only 0.2 percentage points rather than the 4 percentage-point increase that actually occurred. I recommend using “percentage points” instead. Looks like they do the same thing on lines 408, 409, and 427, but that’s harder to confirm from bar graph values.

On line 417, I recommend changing “from” to “relative to/compared to/over” I think the current “due to” works ok, but better would be “from/resulting from”

In the section “Uncertainty in Stream Temperature Projections” beginning at line 505, there should be some mention of uncertainty due to potential impacts of climate change on tree growth, either directly through increases in air temperature and changes in precipitation (and increases in CO2) or indirectly through changes in fire regimes at different elevations and aspects.

689: uncap “Spring-run”

6. PLOS authors have the option to publish the peer review history of their article (what does this mean?). If published, this will include your full peer review and any attached files.

Reviewer #1: No

Reviewer #2: **Yes: **Jon Honea

---

## [Author Response · Author response to Decision Letter 0]

20 Feb 2022

Dear PLOS ONE Editor, 

We would like to thank both reviewers for their time in conducting very helpful reviews of our manuscript. We have responded to the comments from each reviewer below, and have made substantial revisions to the manuscript based on these comments that we believe have significantly improved the manuscript. Of particular note, both reviewers identified a lack of description of the HARP and Chehalis Thermalscape models, which were unpublished at the time this manuscript was first reviewed. We have added description for both models to the methods section of our manuscript, as well as text explaining how our model is distinct from, and builds upon, these two models. Additionally, Beechie et al. 2021 and Jorgensen et al. 2021, which describe the HARP model framework are both now published and publicly available. Please find our responses to the comments from both reviewers below 

Reviewer #1: I reviewed the Research Article manuscript PONE-D-21-16705 entitled “How riparian and floodplain restoration modify the effects of increasing temperature on adult salmon spawner abundance in the Chehalis, River, WA”. This study quantified how much two restoration actions, floodplain restoration and riparian restoration, would reduce stream temperature relative to no-action under climate change scenarios. They then evaluated future habitat conditions and salmonid capacity in the Chehalis River basin for a variety of scenarios (historical thermal conditions, current thermal conditions, future thermal conditions with no-action, future thermal conditions with one restoration action, future thermal conditions with both restoration actions). They find that implementing both restoration actions would result in the greatest decrease in stream temperature, and hence the greatest benefit for salmonids, although specific populations and life stages might benefit more from one restoration action. The amount of work that the authors did is commendable, and this is an important study on how restoration can mitigate negative impacts of climate change for declining and threatened salmonids. However, I cannot currently recommend that this manuscript be accepted into PLOS ONE without major revisions. 

Paper needs major revision before acceptance.

Major problems are present in the Methods and analyses. The authors sometimes based their analyses from other sources with citations (e.g. the HARP and SHaRP frameworks from Beechie et al. 2021 and Jorgensen et al. in review); however, they need to describe these previous results, especially for works that are not publicly available. 

We agree that better descriptions of the HARP model framework was needed to improve the clarity of our methods. The main relevant methods and results of those papers are now included here in the Approach section (paragraph 2). Additionally, we have added a conceptual diagram figure to illustrate the HARP framework (Figure 2). 

I was unclear on how they came up with some of their thermal conversion equations, scaling parameters, and other equations. I also think I found some potential errors in equations as written (e.g. line 326). Background for these needs to be more thorough, or equations need re-analyzed. 

We have addressed the error in the equation on line 326. Additionally, we have modified text in this section to better explain equations and specify where we used equations directly from a source versus where we adapted equations from a source to fit constraints in our analysis (i.e. spring Chinook prespawn holding equation). 

Although unclear, it seems that this manuscript combines the frameworks from Beechie et al. (2021) and Jorgensen et al. in review to assess salmonid capacity under different restoration scenarios. It might be beneficial to 1) publish this as a companion paper, or 2) borrow or adapt a figure from the previous two papers to help illustrate the framework (including equations and parameters borrowed from those papers) more clearly.

The Beechie and Jorgensen papers are now published, so this paper is now essentially a companion paper. We have adjusted the citations for these papers to address the fact that they are now published. Additionally, as requested, we have added a new figure showing a conceptual diagram of how this paper uses the HARP model in order to more clearly illustrate the framework. 

Most other problems are with the writing. The manuscript is too long, is sometimes redundant, is sometimes unclear, and needs some reorganization. 

We have made substantial edits to the manuscript to improve clarity. This included a significant reduction of the text in the uncertainty section as suggested by the reviewer. See responses to other suggested revisions for more details.

There are also too many figures, some of which could be combined, relegated to supplement, or removed. 

Per this suggestion we have moved Figures 2 (ecological regions) and 3 (conversion equations between different temperature metrics) to the supplemental text. 

I provide detailed comments below to help the authors improve their manuscript for future publication. As stated earlier, I think this will be an important contribution showing how to mitigate negative climate change impacts on salmonids. I would be more than happy to review a revised version.

We appreciate these detailed comments and have considered and addressed each comment below. 

ABSTRACT

 General thoughts: It was hard to tell when the authors transitioned from Intro to Methods to Results. It might be more useful to readers to state number of parameters, type of model, and general conclusions in the Abstract.

We have reorganized the abstract to address this comment.

 Line 21: maybe substitute “cold-water” for “anadromous” to point out that an increase in stream temperature is not good for salmonids because they require cold water. You state that your study is on “anadromous salmonid populations” in line 24 – no need to point this out twice in the abstract.

This has been corrected 

 Line 29: “We applied a temperature effect” is vague. Do you mean an temperature effect term in your life cycle or habitat model?

In reworking the abstract, this sentence has been removed. 

 Line 31-32: remove “…and therefore experience the warmest summer temperatures,”

We have removed this statement as suggested. 

 Line 30-32: Unclear here why it is important to note that steelhead are more tolerant of higher temperatures than coho. Is this a result, method, or observation?

This is a result of the temperature-survival functions used in our model. However, we have removed this sentence from the abstract during revisions. 

 Line 38-41: Great sentence for an Abstract. Add ‘%’ to -34

We added in the missing %.

INTRODUCTION

 Line 46: Change “increasing” to “warming”

This has been corrected

 Line 47: Change “increases in sea level” to “rising sea levels”

This has been corrected

 Line 48: Awkward. Sounds like occurrence and intensity of ocean acidity will increase. Rephrase.

We have rephrased this sentence from:

“Climate change is also expected to increase the occurrence and intensity of extreme weather events and ocean acidity, shift global precipitation patterns, and increase regional risk of drought” 

to 

“Climate change is also expected to shift global precipitation patterns, and increase ocean acidity, regional risk of drought, and the occurrence and intensity of extreme weather events”

 Line 52: Remove or relocate “of cultural, economic, and ecological significance”

We have removed “of cultural, economic, and ecological significance”.

 Line 52: Add final sentence here stating something like, “For species already negatively impacted by human activities [Or, for threatened or declining freshwater species], it is imperative to assess how climate change may impact future habitat quality so that…”. Then, start new paragraph after.

We have added the following sentence: 

“Given the threat that human activity poses to many freshwater species, it is essential to understand the impacts that climate change may have on habitat quality in order to effectively manage resilient ecosystems.

We then followed the suggestion to begin a new paragraph following this sentence

Line 56-59: Awkward.

We replaced the existing sentence with the following:

“The projected effects of climate change pose substantial risk to salmonids. These effects include changes to flow regimes resulting from altered patterns in precipitation and snowmelt [12], decreased habitat area due to increased drought, and decreased habitat quality due to warmer than optimum stream temperatures”

 Line 59-61: Don’t discount that warming temperatures could also be beneficial to some life stages, such as juveniles, which grow faster at warm temperatures (to a point) and could exploit warmer habitats. E.g. Armstrong et al. 2021 Nature Climate Change https://www.nature.com/articles/s41558-021-00994-y

We have followed this suggestion and added a sentence here acknowledging the fact that warmer temperatures could also benefit certain salmonid life stages.

 Line 69-71: I think this sentence is the main point of this paragraph – that few salmonid climate change studies quantify how restoration could mitigate negative impacts to populations. Move this idea to the first sentence. May need to rearrange thoughts for flow.

We agree that this idea should be moved up in the paragraph, but in the end felt that it was best for the flow of the paragraph to leave the opening sentence intact. In our opinion our revision to this paragraph preserved a logical order of flow while also highlighting the point that riparian/floodplain restoration for decreasing temperatures is an under-studied area as the reviewer suggested. 

 Line 72-76: Move to preceding paragraph.

We moved these lines to the preceding paragraph as suggested. 

METHODS

 General thoughts: Change the “Study Area/Overview” section to “Study Area and Species”. Shorten this section to description only. Add a new section called “Overview” or “Approach” or similar, where you give a short overview of your approach. This will help the reader to follow your methods logically as you talk about each dataset.

We followed the advice of Reviewer 1 and changed the section titles to “Study Area and Species” and “Approach” and reorganized text as necessary. In the “Approach” section we also added to our descriptions of the HARP model (Beechie et al. 2021/Jorgensen et al. 2021). 

 Line 91-93: Awkward. Maybe, “The Chehalis River Basin consists of 63 subbasins [33,34] and 10 Ecological Regions as defined by…”

We addressed this sentence as suggested.

 Line 103: Need citation.

We added the following parenthetical note for riparian protections in Washington State: (Washington State Salmon Recovery Act was passed in 1999; subsequent administrative rules were enacted in 2001).

Line 135: Simplify “through which we explored” to “to explore”

Addressed. We made the change suggested

 Line 137: “Each future temperature scenario was modeled…”

Addressed. Made the change suggested.

 Line 137: late-century

We addressed this issue and changed “late century” to “late-century” throughout document. 

 Line 142-145: Awkward.

Rephrased to address this comment:

“The restoration scenarios included tree planting and protection within riparian areas to increase stream shading; floodplain reconnection to cool rivers by increasing hyporheic flow; and a combined scenario with both riparian restoration, and floodplain reconnection.”

 Line 146: Change to “Based on reach-level projected stream temperature for each scenario, we estimated…”

We agree that this sentence needed some rephrasing but decided to lead with the estimates rather than the projected stream temperatures. Here is the modified sentence we ended up choosing:

“We modified the life-stage specific fish capacities and productivities based on stream temperature conditions for each scenario, and then used these estimates as inputs to life-cycle models to estimate changes in spawner abundance.”

Line 148-149: You have to give a brief description of the SHaRP model framework here, especially since Jorgensen et al. is still under review.

We agree that the description of the HARP model (formerly SHaRP) was lacking in the manuscript that we submitted. Therefore, we have added more description of this model and its application in Beechie et al. 2021, and Jorgensen et al. 2021 in the “Approach” section of the manuscript. 

Line 149: “the August 7-DADM” is unclear. Do you mean the 7 day period with the warmest 7-DADM in August?

August 7-DADM is defined earlier in the text (lines 97-98 in submitted doc):

”the maximum seven day average of daily maximum stream temperatures for August (7-DADM)”

Therefore, we do not believe it needs to be defined again on line 149. 

 Line 150-152: Simplify to “Our modeled restoration scenarios assumed that…”

Addressed by removing most of the sentence before “assumed that restoration….”. The resulting sentence is as follows:

“Our modeled restoration scenarios assumed that restoration would occur in all reaches with restoration potential (i.e. reaches with degraded riparian condition or disconnected floodplain habitat).”

 Line 157: It might be more readable to divide this into two sections: “Current temperatures” and “Future Temperature Change—No-action”. ----

We have addressed this comment by dividing this section into two sections as suggested

 Line 158-163: Add a brief sentence or two describing the Chehalis Thermalscape spatial stream network model, and its results, since the report is under review and not currently available to readers.

We have added several sentences detailing the Chehalis Thermalscape model and the temporal and spatial covariates it uses to predict stream temperature. (Current Temperatures/Future No-action (Climate Change) Temperatures sections). We reference this as J. Winkowski, unpublished data.

 Line 163-164: Why did you segment a stream into ~200m-long reaches? It seems like it might be better to use 500-1000m-long reaches, to match the stream temperature dataset. Need to address this.

The stream habitat data development and the temperature data development were two different projects. We began the stream habitat analysis in 2016, following the methods of Beechie and Imaki 2013 for segmentation of the stream network (200-m) reaches. This resolution allows assessment of habitat conditions at higher resolution, especially in small streams. At that time we had temperature data from other sources, and Thermalscape was not available. The Thermalscape dataset was developed later by John Winkowski of Washington Department of Fish and Wildlife, using a resolution deemed appropriate for that analysis. We worked together on this manuscript, joining the two datasets for analysis. Ultimately, all habitat attributes in reaches are aggregated to the subbasin level, so the difference in resolution of reaches has little consequence once the data are summarized for the life cycle models.

We modified the following sentence to the text to clarify some of this: 

“For our analysis we segmented the Chehalis Basin river network into reaches approximately 200 m in length, in order to assess a range of habitat conditions at high resolution”

 Line 167: “different” from each other, or different from the stream temp dataset metrics?

We clarified this issue with the following sentence:

“The temperature-survival functions used in our life cycle model require temperature metrics that are specific to the timing of life stages sensitive to temperature”

 Line 167-176: This paragraph is confusing. I understand why the authors need the conversion equations, but I’m unclear why they chose to look at a different temperature dataset with 80 sites of daily stream temp (WDFW Riverscape) to calculate the conversion equations. Were you unable to access the Chehalis Thermalscape project original stream temperature data (presumably, daily temps)? Need a sentence to explain your rationale, or need to calculate conversion equations from the Chehalis Thermalscape project data.

The Riverscape dataset we referred to here was used as input data to the Thermalscape model. We have altered the text to make it clearer that we are using measured temperature data from the same sites used in the Thermalscape model:

“To convert August ADA estimates from the Chehalis Thermalscape dataset to 7-DADM and June 1-21 ADM, we used measured temperatures from the same temperature sensor sites used in the Chehalis Thermalscape model”

 Line 171: need citations and explanations of why each population/life stage function is different.

We have added citations and explanations as suggested:

“Coho and steelhead summer rearing, and spring-run Chinook salmon pre-spawning occur throughout the summer [43,45,44,46], whereas a large percentage of the total spring-run and fall-run Chinook salmon outmigration occurs between June 1 and June 21 [46,48]. Therefore, we used the August 7-DADM in our coho and steelhead summer rearing, and spring-run Chinook salmon pre-spawning temperature-survival functions, and the June 1-21 average of daily maximum temperatures (June 1-21 ADM) for the Chinook salmon outmigration temperature-survival function.”

 Line 184: need citations and short descriptions of the A1B emissions scenario and the VIC model.

We have added citations and descriptions for A1B and VIC (see Future No-action Climate Change Temperatures section)

 Line 187: I thought the Chehalis Thermalscape predicted ADA temps, not 7-DADM. Address.

We agree that this was confusing. The projected increases in 7-DADM were converted from the thermalscape August ADA to 7-DADM in our model. We addressed this by adding “converted to 7-DADM” to clarify that the thermalscape model did not output 7-DADM:

“Projected future increases in stream temperature, converted in our model to 7-DADM…”

Line 191-195: This is precisely how all previous models should be mentioned, with a short description of their model and results, and how you applied it for your study.

 Line 196: Where did you obtain or how did you calculate A, the drainage area?

The drainage area was calculated by creating a flow accumulation raster in ArcGIS (Arc Hydro Toolset). However, during revision we modified the way this equation is presented and removed ‘A’ as a variable. We did not change the calculations themselves, rather, we changed how the equation is presented because we do only use the difference in predicted T from this model, and not the absolute values of T. We now present the following equation, which is how the model is used in this study: 

∆T(growth)= 0.035(∆θ)

 Line 199: Replace “the canopy opening angle” with “”

We were unsure what is being suggested in this comment. We decided to remove “the” and simply say “canopy opening angle”. 

We also reorganized this section a bit to make it more clear why we are using Chehalis Thermalscape as baseline temperature and then using the Seixas et al. model to modify temperature with changes in canopy opening angle. See Future Temperature Change—Stream Shading section for details. 

 Line 199-205: Again, briefly describe lidar, and describe how you defined height class from the lidar data or aerial imagery.

We added the following text to clarify how we analyzed lidar data and aerial imagery to get height class/tree height and then canopy opening angle:

“Tree height was calculated from the lidar dataset as the difference between the elevations of the first returns and the ground surface. Where lidar data were unavailable, we visually estimated canopy opening width and tree size class (Table 2) on each bank at approximately 200m intervals using recent aerial imagery”.

Aerial imagery estimates of height class were visual estimates. 

 Line 208-212: Great idea! Again, briefly describe these growth rates. Are they temperature-dependent or independent? What is the relationship between tree growth, tree height, and ?

The growth rates we were able to find were temperature independent. We added the following sentences for clarity:

“Growth rates for both tree types were rapid at first and leveled off with increasing age. Although future increases in air temperature may impact tree growth rates, the rates used in our study did not vary with differences in air temperature.”

 Line 216: Does the Chehalis Thermalscape include canopy opening angle in its model? Drainage area? How different are the predicted current temps between the Chehalis Thermalscape project and Seixas et al. model? You need to show that Tdiff(growth) is attributed to growth, not differences between the two temp models.

Winkowski and Zimmerman examined a number of potential predictor variables including mean Aug air temperature, mean Aug discharge, drainage area, elevation, channel slope, forest cover, mean annual precipitation, and lake area. However, only mean Aug air temperature, mean Aug discharge, elevation, and mean annual precipiation were significant variables in the Thermalscape model. To account for temperature changes due to shade, we extracted the shade component of the Seixas model and used that to forecast or hindcast temperature change due to differences in riparian tree height. We did not use the actual predicted temperatures from the Seixas model (which predicted lower temperatures on average), only the estimate of change in temperature due to change in canopy opening angle. We felt this combination gave the most accurate estimates of current and future temperatures.

We added more detail about the predictor variables used in the Thermalscape model to the Current Temperatures and the Future No-action (Climate Change) Temperatures sections. Additionally, we added more text clarifying why and how we used the Seixas model to modify temperatures from the Thermalscape model in the Future Temperature Change—Stream Shading section. 

 Line 231-233: Confusing statement. Do the ASRP guidelines dictate that you estimate the widths of floodplain corridors? Clarify.

We re-organized this sentence to make it clear that the guidelines provide the relationship between bankfull and floodplain corridor widths. The guidelines don’t dictate that we must estimate floodplain corridor widths:

“We first estimated the widths of floodplain corridors for each reach based on bankfull width using a relationship outlined in the Aquatic Species Restoration Plan (ASRP) guidelines for the Chehalis River Basin (Table 3)” 

 Line 231 (or so): Somewhere, add a transition phrase: “To define the scaling parameter, we…”

We rephrased this sentence to the following:

“Because proposed floodplain restoration actions in the Chehalis River Basin vary with channel width, we scaled the 2°C projection to each channel width and proposed floodplain width (Table 3).”

 Line 230 on: So, is the scaling factor based on a linear relationship of 0-425 m width and 0-2°C? Is this based on the authors’ assumed estimate or from the study cited above? I am concerned that the authors are defining a linear relationship that may not be linear. Please elaborate, clarify, or revise.

We have added description of the Seedang et al. model in the Future Temperature Change—Floodplain Reconnection section in order to both clarify and justify our use of this model:

“To model alternative restoration scenarios, Seedang et al. [30] created a linear model, relating temperature reduction to the area of connected floodplain features (primarily channels and islands), despite underlying complexities in the estimation of flow path length, hydraulic head, cooling in each flow path, and other factors. While one might expect that temperature reduction will decrease as reconnected floodplains become very wide, we are unaware of other models that may describe a non-linear relationship. Moreover, the floodplain reconnection widths we modeled are narrower than the maximum width modeled in our source study, so we are applying the linear model within a floodplain width range that is consistent with the original model.”

Also:

“While historical floodplain widths often exceed those in the restoration plan, we did not model a temperature effect of wider (historical) floodplains because we were not certain that temperature reduction continues as floodplain width increases beyond those widths of the original study.” 

 Line 235-236: Does it matter to distinguish between large and small river reaches?

We differentiate between large and small river reaches because the floodplain-temperature effect is applied to all large river reaches, as well as any small stream reaches with disconnected marsh habitat, but not small stream reaches with no disconnected marsh habitat. This assumes that all large river reaches in the basin have some amount of disconnected floodplain area. We have altered the text to clarify based on the comment:

“We then applied the linear relationship between restored floodplain width and stream temperature change [30] (Table 3) to estimate temperature change to all large river reaches (> 20 m bankfull width) throughout the basin, as well as any small stream reaches (< 20 m bankfull width) with documented disconnected marsh habitat.”

 Line 242: Use subscripts to make this equation easier to read. Also, need to elaborate on this relationship since it is from another reference.

We have adopted the use of subscripts based on this comment. This relationship was developed using measured daily temperatures from the same sites used in the Chehalis Thermalscape model. We have clarified this in the text. 

 Line 249: Need to briefly describe the SHaRP for your readers. Is that what you do following? Unclear.

See prior response. We have added a section with description of the HARP model in the Approach section.

 Line 251-252: Unclear after “is affected through…” Revise.

We have rephrased this sentence to make it more clear:

“Temperature influences habitat quality, which impacts both density independent survival (productivity) and capacity, defined as the number of individuals that can be sustained by a particular habitat.”

We then clarify that capacity = area * density (removed from the sentence above) later in in the section.

 Line 261-266: When borrowing estimates from other sources, briefly describe their work so that readers know a bit about those estimates. For example, “[33] combined geospatial layers, recent field surveys, and models to estimate historical and current habitat conditions in the Chehalis River basin.” Elaborate on your capacity and productivity multipliers.

We added the following text to clarify how estimates were obtained:

“Estimates of current habitat area and condition for each subpopulation were obtained through geospatial analysis and modeling [35], whereas density and productivity estimates were drawn from field surveys and values from published literature (see Jorgensen et al. [36]). Life stage capacity was calculated as the product of habitat area and life stage- and species-specific density [36]. “ 

We also added the following text to clarify how capacity and productivity multipliers are used:

“For each species and life stage impacted by temperature, we developed a temperature multiplier based on documented relationships between stream temperatures and salmonid capacity and productivity. We then used these multipliers to scale the capacity and productivity parameters used in the life cycle models (described below).”

 Line 279: Remove “based on the 7-DADM stream temperature”

We removed this as suggested

 Line 277 on: Combine the temperature multipliers sections into a single section for clarity. 

 Give a short introduction as to what the temperature multiplier is for,

 then talk about your multiplier for each life stage/population in separate paragraphs. 

 Once you do this, you can reduce a lot of redundant text.

As suggested, we combined the temperature multipliers sections into a single section. Additionally, we added text to clarify the purpose of the temperature multiplier in the Modeling Effects of Temperature Change on Spawner Abundance section preceding the Temperature Multipliers section. 

 Line 308-311: Rephrase to something like, “Although juvenile Chinook salmon can outmigrate from xx through xx, 45% of parr oumigrate between June 1-21 [45].” This keeps the focus that you are still estimating thermal habitat during the peak of the outmigration.

We have rephrased this section as suggested. 

 Line 326: Check this equation.

 First, it is an odd way to write this, making me think there is an error. 

There was indeed an error, and we have fixed the equation to remove the error.

 Second, I cannot find this equation in ref 55. 

The equation is adapted from Bowerman, not directly from it. Bowerman et al. include hatchery origin spawners, whereas we removed these fish from the function for our analysis. We added “adapted” to the text to make it clearer that we changed the function slightly from what is presented in Bowerman. Additionally, we added text describing how we adapted the function (removal of hatchery fish).

 Third, plugging in temperatures from 12-24 C makes the temp_multiplier a maximum of 0.006. Is that correct??

The low temp_multiplier maximum was due to an error in the equation as the reviewer identified. This is now fixed, and the max temp multiplier is roughly 1 as expected. 

RESULTS

 Line 331: Here or somewhere in the Methods, add a sentence to explain why you are evaluating these threshold categories. This could be something like: “To be protective of all salmonid life stages and populations, we defined temperatures above 24C as unsuitable.” Or, “Temperatures <18C are generally good for all life stages.”

We added the following sentence to clarify this:

In our analysis we considered stream temperatures >24°C 7-DADM (lower limit of UILT for juvenile salmonids) to be generally unsuitable for most populations, whereas we considered temperatures <18° to be generally suitable.

 The ’Temperature Change’ section is very long. Condense. From my view, your results are essentially: 1) Temperature is predicted to increase with climate change. 2) Each restoration scenario reduces the temperature relative to no action. 3) Combining restoration scenarios produced the greatest reduction in modeled stream temperature. 4) Restoration actions will still result in warmer temperatures.

We decided to take out percent changes in the number of reaches within different temperature bins that were already expressed in Table 5. This cut down on some of the unnecessary text in this section. 

 Line 393-399: Nicely written.

 Line 406: indicated

Changed “indicate” to “indicated” as suggested

 Line 412-442: Verbose. Condense.

We condensed this section as suggested.

DISCUSSION

 I prefer to see the first sentence of the Discussion as a short summary of what you did and why. Then, lead with your most important result or conclusion. Is your most important result that salmonid populations will be impacted differently by climate change?

Following the advice of the reviewer we used what was the first paragraph of the conclusion to the first paragraph of the discussion. We agree that this is a good fit. We then moved the first paragraph of the discussion to the first paragraph of the conclusion. 

 Line 449-451: Awkward

We rephrased this section to the following:

“Our model suggests that restoration opportunities exist within the Chehalis River Basin that could mitigate climate change-induced declines for all four populations, and that sustaining vulnerable salmonid populations in the Chehalis River Basin into the future will likely require a management approach that prioritizes limiting increases in stream temperature.”

 Line 452-465: nicely written

 Line 469-470: Move the phrase after “whereas the opposite…” to ~line 475 “On the other hand…”

We reorganized the first few paragraphs of the discussion section as suggested below. We now have separate paragraphs discussing benefits of riparian restoration and floodplain reconnection for each population. This comment has been addressed through this reorganization. 

 Line 480-493: First, juvenile outmigration here refers only to parr, correct? What about yearlings that rear over summer? Second, why the focus on this paragraph on only Chinook and not coho and steelhead as well?

Chinook salmon juvenile outmigration in the HARP model is broken into fry and subyearling migrants (Jorgensen et al. 2021, Identifying the potential of salmon habitat restoration with life cycle models). In the Chehalis Basin, very few Chinook rear over summer (i.e., there are no known yearling migrants in this basin) therefore we do not consider yearling chinook migrants in our analysis. 

The focus of this paragraph was a comparison of the most and least affected populations in our model (Spring/Fall Chinook). However, during reorganization of the first part of the discussion we ended up deleting this paragraph as most of this was already covered:

 lines 480-485 are essentially covered in methods

 Lines 487-490 also essentially covered in methods

 Lines 490-493 covered later in discussion

 Line 484-485: Maybe – but June is peak outmigration time, right? So there might be a benefit to entering the ocean then compared to earlier in the year (maybe juveniles are larger and can more easily withstand predation (e.g. Beamish and Mahken 2001), or maybe juvenile prey is more abundant then). If this is the case, a negative impact during peak timing is still damaging to the population, right?

See comment above. During reorganization of the first part of the discussion we ended up deleting this paragraph as most of this was already covered:

 lines 480-485 are essentially covered in methods

 Lines 487-490 also essentially covered in methods

 Lines 490-493 covered later in discussion

 It might be beneficial to reorganize a few paragraphs in the Discussion to avoid redundancy and to increase flow and clarity. My views are: 1) Short summary paragraph of most important results. 2) Leave second paragraph as is. 3) Keep the focus on restoration actions. “Restoration actions were predicted to impact salmonid populations differently, dependent on freshwater habitat use and life history characteristics.” Then something like, “Riparian shading would greatly benefit salmonids over-summering in small streams.” Then discuss which populations riparian shading helps and why. 4) “On the other hand, floodplain restoration was predicted to strongly benefit salmonids oumtigrating or rearing during the summer in large rivers. Juvenile steelhead and Chinook salmon…” 5) “We found that implementing both restoration actions would result in the lowest increases in stream temperature relative to no-action. Implementing both actions is particularly important for spring-run…”

We made substantial revisions here following the reorganization suggested by the reviewer. We removed the final paragraph (see lines 480-493 in original doc) because everything is stated elsewhere and the paragraph felt out of place. This eliminated the sections addressed in comments above about lines 480-493 and 484-485. 

 Line 506-521: This can be condensed. It’s OK that you did not model every future temperature scenario. Simply state that other air temp projections could result in cooler or warmer stream temperatures than the A1B scenario used in [30]. Uncertainty in stream temp projections: Can you condense the first three paragraphs into one? They seem to be making the same point regarding uncertainty in air temp and flow projections.

Following the suggestion of the reviewer we have significantly condensed our model uncertainty sections, as many of the details included here can be found elsewhere in the paper, or were more detailed than was necessary to include. This also addresses the reviewer’s comment that our paper was too long and needed condensing. 

 Line 583: change “are” to “were here”

We made the change suggested here

 Line 589 (or so): Additionally, other factors can change these thermal relationships. For example, food availability, flow velocity, or predator density. Add a sentence or two here describing these impacts on thermal tolerances, and implications for your results.

We discuss these factors in the following paragraphs of the discussion, therefore we have chosen not to adopt this suggestion in order to avoid redundancy. 

 Line 590-606: So, are you saying that only fall-run Chinook have the potential to change their phenology to avoid climate change? Why or why not?

We are not saying that only fall chinook may change phenology. Rather, we are giving this as an example of one way in which a population may adapt to avoid climate change, and why this doesn’t necessarily make our model invalid. We added text to clarify that this is just an example of a population that may change phenology. 

 Line 607-609: This seems like it should fit better in the “Uncertainty in stream temp projections” section. Maybe change section name from projections to modeling. Then elaborate. The rest of the paragraph is probably fine here.

We believe that this paragraph should stay together but agree that this concept is more related to stream temperature projections/modeling than to salmon response to temperature. Therefore, we moved the whole paragraph up to the Uncertainty in Stream Temperature Projections and Modeling section and changed the name of the section. 

 Line 637-638: Awkward phrasing. Maybe simplify to “An increase in thermal tolerance by 1-3C could boost coho…”

We rephrased this section to the following: 

“We found that an increase in thermal tolerance of 1-3°C could boost coho spawner abundance in our temperature scenarios by as much as 7-31%.”

 Line 640: as above.

We rephrased this section to the following: 

“On the other hand, a reduction in thermal tolerance of 1-3°C could decrease coho spawner abundance in our temperature scenarios by as much as 8-23%.” 

 Line 633-657: Interesting ideas, to increase thermal tolerance by increasing prey availability, and thereby help salmonids cope with climate change. I think there might also be research regarding increasing prey availability on floodplains. Add a citation/sentence if you have one handy.

We have included several citations to address this comment

 Recycle the “Other Considerations” sentences to other sections. Condense.

In the other considerations section there were some considerations related to temperature and some related to other aspects of climate change. We opted to move those associated with temperature into the previous section and keep those not directly related to temperature (ocean conditions, non-native species, etc.) in the other considerations section for consistency. Through this process we were able to condense a bit. 

 Line 684-696: I think this is a great first paragraph of your Discussion.

Following the advice of the reviewer we moved what was the first paragraph of the conclusion to the first paragraph of the discussion. We agree that this is a good fit. We then moved the first paragraph of the discussion to the first paragraph of the conclusion. 

 Good final paragraph.

 Throughout your Discussion, you find that spring-run Chinook salmon are particularly vulnerable. This result is also replicated in other studies. Start with FitzGerald et al. 2021 Global Change Biology and references within. Add to your Discussion to bolster the results of your study.

We appreciate the suggestion and have added citations for Fitzgerald and Crozier papers in first paragraph of the discussion

REFERENCES

 Ref. 33: Update

 Ref 49: Not all info included.

We have updated both references 33 and 49 (reference number has changed in the updated draft). 

FIGURES AND TABLES

 Table 1: nice table. This is not necessary, but it may strengthen your paper if you have hypotheses for each scenario. For example, “Riparian restoration” is hypothesized to cool temperatures in the summer, which helps salmonids present in the summer (i.e. Chinook spring-run pre-spawn adults, juvenile coho, juvenile steelhead), but may not impact fall-run Chinook. I think most of this information is present in your Study Area/Overview section.

As the reviewer noted, much of this information can be found in the text. Therefore, we adjusted the table caption to reflect the fact that hypotheses can be found within the text

 Table 4: Put spring-run and fall-run Chinook salmon on separate rows. You repeatedly talk about the four salmonid populations, so keep that cohesive. Then, you can remove the footmark.

We made these changes as suggested

 Table 5: This table might present better as a figure. For example, a bar plot showing how % for each temp category changes from natural potential to current to mid-century to late-century, with each restoration action as different groupings.

We attempted to present these results in a figure but found that presentation in a table was cleaner than in a figure. We opted to keep Table 5 as is. 

 Fig.1: nice figure. The yellow for agriculture gets lost a bit, even at high res – maybe brighten to orange, or darken the green for forest. Add ‘Columbia River’ to inset map for reference to those not familiar with the region. Is the forest/agriculture/developed characteristics important for your model, or just as interesting background?

We darkened the green and yellow in this figure so that both show up better. We also added a note about the Columbia River in the inset map in the caption. Forest/agriculture/developed characteristics do not directly influence our model, although they are relevant to the restoration potential of a reach. 

 Fig. 2: nice figure, but is this necessary for your model? Might be better as a supp fig.

We agree and will move this figure to the supplementary text

 Fig. 3: Change y axis to “August 7-DADM” for clarity. Add (°C) to all axes.

We have added °C to axes and changed the y axis to August 7-DADM as suggested. We have also decided to move this figure to the supplementary text in order to reduce the number of figures. 

 Fig. 4 legend: Check your reference numbers here. Are these really the best references for these thermal performance curves for each species?

We added some more detail, but these are the references used to create these curves. We added more detail to the text of the caption to make this clearer.

Fig. 5. Very nice figure.

 There might be a way to condense Fig. 5-7 into a single figure. Essentially, a multi paneled map fig with rows vs. columns as time vs. restoration actions. You could show predicted temperature or predicted temperature changes and still get your point across more simply.

We seriously considered combining figures 5-7 into a single figure, however we couldn’t figure out a way to combine them without losing information or making the maps too small. Therefore, we decided to keep figures 5-7 as three separate figures. 

 Fig. 8 legend: Need citation for estimated spawner abundances.

These “estimates” were produced by our model. So we changed “estimated” to “modeled” to clarify that these are from our efforts and no citation is needed. 

 Fig. 8: nice figure.

 Fig. 9: Why do you show spawner abundance by ecological region rather than along stream reaches? It might be more helpful to show calculated values (spawners per reach), or show % of maximum spawners in entire basin for each reach. Also, what is the gray?

Spawner abundance is calculated at the subbasin level in our life cycle model, therefore we don’t actually have reach level spawner/change in spawner estimates. 

We added the following sentence to the figure caption regarding gray regions: “Gray regions are outside of the spawning and rearing distributions for a given population.”

SUPPLEMENTARY FIGURES

 These are the figures (S2-S4) I was thinking about earlier with combining Figs. 5-7 into a new simpler figure. Think about if you could combine them. If you cannot, think about if you need legends for the supp figs.

Reviewer 2

Reviewer #2: Fogel et al.’s “How riparian and floodplain restoration modify the effects of increasing temperature on adult salmon spawner abundance in the Chehalis River, WA” describes a multi-model exercise to evaluate estimated impacts of future water temperature increase and habitat restoration on 4 salmonid populations that reproduce in the Chehalis River. They find that modeled increases in water temperature due to climate change will reduce spawner numbers of all four populations at both mid-century and late century. However, modeled habitat restoration shows the potential to at least partially ameliorate those impacts. In particular, modeled restoration of riparian forests has the potential to benefit life stages occupying smaller streams and modeled restoration of floodplain connectivity has the potential to benefit life stages occupying wider channels such as the main stem.

The lack of availability of some important details of their work means I cannot fully evaluate this manuscript. At least three key sources describing their life-cycle models as well as their main stream temperature model are under review and weren’t made available to me. The life-cycle models estimating salmon spawner abundance as influenced by stream temperature are described in Beechie et al. and Jorgensen et al., citations 33 and 34, both under review at PLoS ONE. Likewise, the details of their main stream temperature model, the Chehalis Thermalscape spatial stream network temperature model, can be found in Winkowski and Zimmerman, citation 30, which is under review at the Washington Department of Fish and Wildlife.

We appreciate the detailed comments provided by the reviewer and have considered and addressed each comment below. 

Beechie et al. 2021 and Jorgensen et al. 2021 are now published and publicly available. However, we did add more detail on the HARP model framework in the “Approach” section of this manuscript during our revisions. 

The new manuscript for the Chehalis Thermalscape model (Winkowski, In Review) is still in review, however we have added a citation for a previous version of the Chehalis Thermalscape model so that readers are able to better understand this model. Additionally, we added more details on the methods used in this model in the Current Temperatures and Future No-action (Climate Change) Temperatures sections within the Methods section.

Winkowski J, Zimmerman M. Thermally suitable habitat for juvenile salmonids and resident trout under current and climate change scenarios in the Chehalis River, WA. Olympia, WA: Washington Department of FIsh and Wildlife; 2019. resident

If those works pass review without substantial changes, there remains the question of whether the models they elaborate were used appropriately here. For example, I would like to see how productivity and capacity are used to determine the number of fish surviving each life history stage. Line 289 indicates that a temperature multiplier was applied to both productivity and capacity. This is different from some life-cycle models where that would be appropriate only for productivity. 

Throughout the development of the HARP model we had input from a science review team with members experienced with life cycle models. It was their recommendation that based on the math that goes into the life cycle models, this function must affect capacity and productivity. 

That said, if the three sources describing their key models pass review and if they are used appropriately here, then this work is an important contribution to the management and restoration of salmonid populations in the face of climate change. I see no need for the authors to run additional model scenarios. However, I do recommend below some changes to provide important clarification.

See above, two of the three sources mentioned are now publicly available and we have added an additional citation for a previous version of the third source

The authors indicate on line 185 that the estimates of future water temperature and stream flow produced by the water temperature–climate model they used were based on the A1B emissions scenario. I recommend adding a sentence explaining why the A1 storyline and scenario family and why the B group of the A1 family are appropriate for this exercise and what the use of A1B assumes about the global carbon emissions pathway. Comparing A1B to RCP scenarios 4.5 and 8.5 beginning at line 518 is useful, but there too an indication of what the RCP scenarios assume about carbon emissions pathways would be useful context.

We have added in a brief description of the A1B and RCP scenarios. The A1B description can be found in the Future No-action (Climate Change) Temperatures section of the Methods. The RCP description can also be found in the Future No-action (Climate Change) Temperatures section. 

186: recommend specifying “water” temperature (or “air” if that’s what they mean here)

The sentence structure has changed during reorganization we have made sure to clarify where we are referring to water vs air temperature in this and other sections

At line 230, the authors describe using water temperature responses to riparian reconnection in a wider channel section of the Willamette River to scale responses for narrower channel sections on the Chehalis, producing Table 3. They should explain how they produced a linear scale from just one point (425, 2) or indicate that more than one width-temperature point was available from the Willamette.

We have added the following text to clarify how and why we are using this relationship:

“One study on the Willamette River projected that reconnecting floodplain features within a floodplain corridor roughly 425 m wide would decrease 7-DADM temperature by 2°C [30]. To model alternative restoration scenarios, Seedang et al. [30] created a linear model, relating temperature reduction to the area of connected floodplain features (primarily channels and islands), despite underlying complexities in the estimation of flow path length, hydraulic head, cooling in each flow path, and other factors. While one might expect that temperature reduction will decrease as reconnected floodplains become very wide, we are unaware of other models that may describe a non-linear relationship. Moreover, the floodplain reconnection widths we modeled are narrower than the maximum width modeled in our source study, so we are applying the linear model within a floodplain width range that is consistent with the original model [30]”

“While historical floodplain widths often exceed those in the restoration plan, we did not model a temperature effect of wider (historical) floodplains because we were not certain that temperature reduction continues as floodplain width increases beyond those widths of the original study. Hence, the modeled temperature reductions in Table 3 also represent the maximum potential temperature reduction for floodplain reconnection in the model.” 

This reader was confused about the mention of a multiplier at line 256 without further elaboration until line 278. I recommend adding something like “described below” to the end of the sentence on line 256.

We added more detail about the temperature multipliers in this sentence and also added “described below” to the end as suggested. 

The use of % is ambiguous on Lines 332–338, 362, 364, 375, 375, 387, 389, and 390. For example, beginning on line 331, they state that “the number…increased from the current scenario by 4%.” However, a 4% increase of 5% might mean an increase of only 0.2 percentage points rather than the 4 percentage-point increase that actually occurred. I recommend using “percentage points” instead. Looks like they do the same thing on lines 408, 409, and 427, but that’s harder to confirm from bar graph values.

We condensed the results section to reduce redundancy, so many of the percentages mentioned by the reviewer are no longer in the text. Instead, Table 5 lists these results. 

On line 417, I recommend changing “from” to “relative to/compared to/over” I think the current “due to” works ok, but better would be “from/resulting from”

We made the changes suggested by the reviewer here

In the section “Uncertainty in Stream Temperature Projections” beginning at line 505, there should be some mention of uncertainty due to potential impacts of climate change on tree growth, either directly through increases in air temperature and changes in precipitation (and increases in CO2) or indirectly through changes in fire regimes at different elevations and aspects.

We have added text specifying that although changes to environmental conditions may impact tree growth we do not vary tree growth rates based on changes to these conditions in our model. The added text can be found in the Future Temperature Change—Stream Shading section. 

689: uncap “Spring-run”

We made this change in the text

---

## [Editor Report · Decision Letter 1]

10 May 2022

How riparian and floodplain restoration modify the effects of increasing temperature on adult salmon spawner abundance in the Chehalis, River, WA

PONE-D-21-16705R1

Dear Dr. Fogel,

We’re pleased to inform you that your manuscript has been judged scientifically suitable for publication and will be formally accepted for publication once it meets all outstanding technical requirements.

Kind regards,

Rachel A Hovel

Academic Editor

PLOS ONE

Additional Editor Comments (optional):

I do have several comments to be addressed in your final submission. 

1. The introduction starts very broadly, well outside of the scope of your study (e.g. mentions of sea ice and ocean acidification). I suggest trimming this tangential material. 

2. The methods subheading "Approach" lacks descriptive clarity (replace with, e.g. "Temperature and habitat modeling framework"). 

3. Some inconsistencies in font exist in the Discussion. 
---

## [Editor Report · Acceptance letter]

1 Jun 2022

PONE-D-21-16705R1 

How riparian and floodplain restoration modify the effects of increasing temperature on adult salmon spawner abundance in the Chehalis River, WA 

Dear Dr. Fogel:

I'm pleased to inform you that your manuscript has been deemed suitable for publication in PLOS ONE. Congratulations! Your manuscript is now with our production department. 

Kind regards, 

on behalf of

Dr. Rachel A Hovel 

Academic Editor

PLOS ONE